# MiAD: Mirage Atom Diffusion for De Novo Crystal Generation

## Abstract

In recent years, diffusion-based models have demonstrated exceptional performance in searching for simultaneously stable, unique, and novel (S.U.N.) crystalline materials. However, most of these models don't have the ability to change the number of atoms in the crystal during the generation process, which limits the variability of model sampling trajectories. In this paper, we demonstrate the severity of this restriction and introduce a simple yet powerful technique, mirage infusion, which enables diffusion models to change the state of the atoms that make up the crystal from existent to non-existent (mirage) and vice versa. We show that this technique improves model quality by up to $\times 2.5$ compared to the same model without this modification. The resulting model, Mirage Atom Diffusion (MiAD), is an equivariant joint diffusion model for de novo crystal generation that is capable of altering the number of atoms during the generation process. MiAD achieves an $8.2\%$ S.U.N. rate on the MP-20 dataset, which substantially exceeds existing state-of-the-art approaches.

## 1 Introduction

Deep generative modeling opens new horizons of possibilities across various fields. Its impact is particularly profound in the natural sciences, where generative models can act as powerful exploration engines (Jumper et al., 2021) capable of dramatically accelerating scientific and technological progress. One area where this potential is especially evident is materials science. Despite the vast number of hypothetically possible materials, only a small fraction are capable of stable existence under physical constraints, the underlying principles of which remain largely unknown; this poses a significant challenge to the discovery of new materials.

Several approaches to deep generative modeling, including Generative Adversarial Networks (GANs) (Goodfellow et al., 2014; Karras et al., 2019), Variational Autoencoders (VAEs) (Kingma & Welling, 2013), Large Language Models (LLMs) (Devlin et al., 2019; Touvron et al., 2023), as well as Diffusion models (Sohl-Dickstein et al., 2015; Ho et al., 2020; Song et al., 2020; Vahdat et al., 2021) and Flow-Matching models (Lipman et al., 2022) shown strong performance in generating complex objects. The latter two paradigms have demonstrated their potential in handling the trade-off between object quality and generation diversity (Xiao et al., 2022). These models are the same in organizing the generation process as a trainable, iterative process, which progressively refines the output over a sequence of steps.

Applying diffusion models to data from the natural sciences is non-trivial due to the need to encapsulate specific properties within a trainable dynamic framework. Multiple approaches are tailored for this objective, each addressing a particular class of properties. Currently, diffusion models can operate in non-Euclidean spaces (Huang et al., 2022; De Bortoli et al., 2022; Chen & Lipman, 2023; Okhotin et al., 2023; Hoogeboom et al., 2021) and exhibit symmetrical properties (Hoogeboom et al., 2022; Klein et al., 2023).

In recent years, diffusion models have demonstrated their potential in the domain of material science. The DiffCSP model (Jiao et al., 2024a) has shown ability to generate 3D crystal structures while effectively capturing the intrinsic regularities of the crystal manifold. Additionally, the MatterGen model (Zeni et al., 2024) demonstrated that diffusion models can generate stable and novel materials – an essential quality characteristic in this task (Kazeev et al., 2025). While it is natural to expect generative models to generalize beyond the training data and create novel samples, this objective

can sometimes conflict with the training loss, which often emphasizes fidelity to the observed data distribution. Such tension can negatively influence the model's ability to explore new regions of the data space.

However, by design, these models are restricted to generating crystals with a fixed number of atoms, which limits their ability to perform intuitive operations such as adding or removing specific atoms during generation. As we show in Section 6, this constraint reduces the model's flexibility and hinders its ability to explore a broader range of plausible crystal structures, impacting both the diversity and quality of the generated outputs.

In this paper, we present the next steps toward improving diffusion models for crystal generation and demonstrate their significance in the search for novel and stable materials:

- We propose a simple yet powerful technique, *mirage infusion*, which broadens the original space of 3D crystal structures, enabling the diffusion model to modify the number of atoms in the crystal during the generation process.
- We examine the sensitive parameters of the proposed technique and their impact on the quality of the generative model through a series of experiments.
- We demonstrate that the proposed approach significantly enhances the performance of the base joint diffusion model and substantially surpasses the previous state-of-the-art model.

## 2  PRELIMINARIES

### 2.1  DIFFUSION MODELS

A diffusion model is a generative model that consists of forward and backward processes:

$$q(x_{0:T}) = q(x_0) \prod_{t=1}^{T} q(x_t|x_{t-1}), \quad p_\theta(x_{0:T}) = p(x_T) \prod_{t=1}^{T} p_\theta(x_{t-1}|x_t),$$

where $q(x_0)$ represents the data distribution, $q(x_t|x_{t-1})$ denotes the transition kernel which gradually adds noise to data objects, $p(x_T)$ is the prior distribution from which the diffusion model begins generation, and $p_\theta(x_{t-1}|x_t)$ is the trainable transition kernel used for step-by-step object denoising during the sampling procedure. The original training objective for the backward process is the Evidence Lower Bound (ELBO). Alternatively, in continuous spaces, diffusion models can be reformulated using *score-matching*. In this framework, the model is trained to approximate the *conditional score* $\nabla_{x_t} \log q(x_t|x_0)$, while sampling relies on the *unconditional score* $\nabla_{x_t} \log q(x_t)$.

Finally, for data with complex internal structures, one can factorize objects into several components and define a separate diffusion process for each of them. As an example, crystals can be factorized into three components: lattice matrix, fractional coordinates of atoms and atom types, where each component belongs to a space with a specific geometry that differs from the others. In the model, which we extend in this work, the crystal structure is decomposed into a lattice matrix, fractional atomic coordinates, and atom types — each residing in a space with distinct geometric properties. In this setting, the sampling procedure requires simultaneous denoising of all these components. Therefore, we can use a single neural network, which takes the noisy versions of all object components and denoises them according to their respective diffusion processes. In this way, the diffusion model for each component is conditioned on the noisy versions of all object components forming a *joint diffusion model*. The resulting training objective is a weighted sum of the objectives from all diffusion processes.

### 2.2  CRYSTAL REPRESENTATION

The representation of a 3D crystal can be reduced to the representation of a *unit cell*, with the entire crystal being an infinite repetition of its unit cell. A unit cell is represented by the lattice $L = [l_1, l_2, l_3] \in \mathbb{R}^{3 \times 3}$ — three basis vectors that define its geometry, $F \in [0, 1)^{N_{\text{atoms}} \times 3}$ — the fractional coordinates of atoms within the unit cell, and $A \in \{1, \ldots, N_{\text{types}}\}^{N_{\text{atoms}}}$ — the types of atoms. The number of atoms in the unit cell, $N_{\text{atoms}} \in \mathbb{N}_+$, varies depending on the specific crystal.

The entire crystal is an infinite periodic structure, defined as a set of atoms given by:

$$\{(a_i, x_i) \mid x_i = (f_i + k)L, \ \forall k \in \mathbb{Z}^3\},$$

where each atom is represented by a pair consisting of its type $a_i$ and its Cartesian coordinates $x_i$. Thus, the task of generating a crystal reduces to generating the triplet $\mathcal{M} = (L, F, A)$.

## 2.3 DIFFUSION MODEL FOR CRYSTALS

The joint diffusion model over crystal structures $\mathcal{M} = (L, F, A)$ is proposed in Jiao et al. (2024a); Zeni et al. (2024). In this model, the forward process is factorized across the lattice $L$, fractional coordinates $F$, and atoms types $A$:

$$q(\mathcal{M}_t | \mathcal{M}_{t-1}) = q(L_t | L_{t-1}) q(F_t | F_{t-1}) q(A_t | A_{t-1}),$$

where $\mathcal{M}_0 = (L_0, F_0, A_0)$ denotes a clean crystal, while $\mathcal{M}_t = (L_t, F_t, A_t)$ represents an intermediate noisy version of the crystal.

### 2.3.1 DIFFUSION COMPONENTS

**Lattice** Given that $L_0 \in \mathbb{R}^{3 \times 3}$, we can employ DDPM (Ho et al., 2020) with Gaussian distributions:

$$q(L_t | L_{t-1}) = \mathcal{N}\left(L_t | \sqrt{\beta_t} L_{t-1}, \sqrt{1 - \beta_t} I\right)$$

$$p(L_T) = \mathcal{N}\left(L_T | 0, I\right)$$

$$p_\theta(L_{t-1} | \mathcal{M}_t) = \mathcal{N}\left(L_{t-1} | \mu_\theta(\mathcal{M}_t, t), \sigma_t^2 I\right),$$

where $\beta_t \in \mathbb{R}_+$ – sets the scale of noise for step $t$ and $p_\theta(L_{t-1} | \mathcal{M}_t)$ is conditioned on the noisy versions of all crystal components, as it is a joint diffusion model. The training objective for this diffusion component is defined as a weighted ELBO, which reduces to the following form:

$$\mathcal{L}_L = \sum_{t=2}^{T} \gamma_t \mathbb{E}_{\mathcal{M}_0 \sim q(\mathcal{M}_0), \mathcal{M}_t \sim q(\mathcal{M}_t | \mathcal{M}_0)} D_{\mathrm{KL}}\left[q(L_{t-1} | L_t, L_0) \,||\, p_\theta(L_{t-1} | \mathcal{M}_t)\right]$$

**Fractional coordinates** The space of fractional coordinates is periodic. Therefore, we need to utilize a diffusion model that can accommodate data of this nature. One possible option is to use a Wrapped Normal distribution for each coordinate of each atom:

$$q(F_t | F_{t-1}) = \mathcal{WN}\left(F_t | F_{t-1}, (\sigma_t^2 - \sigma_{t-1}^2)I\right)$$

$$p(F_T) = \mathcal{U}\left(F_T | 0, 1\right),$$

where $\sigma_t \in \mathbb{R}_+$ – set the scale of noise for step $t$. A particularly suitable and efficient method for training the backward process in this model is to adopt Riemannian score matching (De Bortoli et al., 2022) with the following objective:

$$\mathcal{L}_F = \mathbb{E}_{\mathcal{M}_0 \sim q(\mathcal{M}_0), t \sim \mathcal{U}(1,T), \mathcal{M}_t \sim q(\mathcal{M}_t | \mathcal{M}_0)} || \nabla_{F_t} \log q(F_t | F_0) - s_\theta(\mathcal{M}_t, t) ||_2^2,$$

where $s_\theta$ is a neural network that approximates the unconditional score for the generating procedure proposed by Jiao et al. (2024a)

**Atom types** The atom type is a discrete variable drawn from a fixed set of $N_{\text{types}}$ possible elements. We use D3PM (Austin et al., 2023) for this component:

$$q(A_{t,i} | A_{t-1,i}) = \mathrm{Cat}\left(A_{t,i} | Q_t A_{t-1,i}^{\text{onehot}}\right)$$

$$p(A_{T,i}) = \mathrm{Cat}\left(A_{T,i} | \mathbf{1} / N_{\text{types}}\right)$$

$$p_\theta(A_{t-1,i} | \mathcal{M}_t) = \mathrm{Cat}\left(A_{t-1,i} | c_{\theta,i}(\mathcal{M}_t, t)\right),$$

where $A_{t,i}$ – the atom $i$ in the crystal $\mathcal{M}_t$, $A_{t-1,i}^{\text{onehot}}$ – zero vector of size $N_{\text{types}}$ with 1 on position $A_{t-1,i}$, $\mathbf{1}$ – unit vector of size $N_{\text{types}}$, $Q_t \in \mathbb{R}_+^{N_{\text{types}} \times N_{\text{types}}}$ – matrix that gradually spreads probability

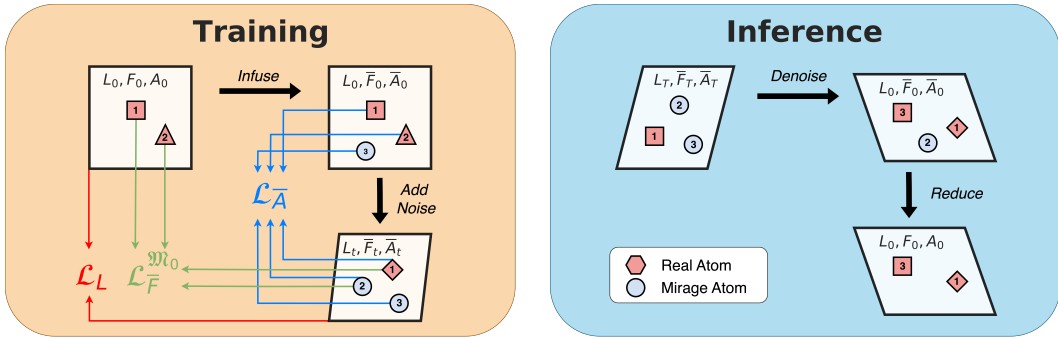

Figure 1: Overview of the proposed mirage infusion technique.

mass across all types on each step $t$, $c_{\theta,i}$ – prediction of probabilities of each type for atom $i$. The training objective takes the following form:

$$\mathcal{L}_A = \sum_{t=2}^{T} \mathbb{E}_{\mathcal{M}_0 \sim q(\mathcal{M}_0), \mathcal{M}_t \sim q(\mathcal{M}_t|\mathcal{M}_0)} \sum_{i=1}^{N_{\text{atoms}}} D_{\text{KL}}\left[ q(A_{t-1,i}|A_{t,i}, A_{0,i}) \,||\, p_\theta(A_{t-1,i}|\mathcal{M}_t) \right]$$

Although the model is capable of generating crystals with different numbers of atoms, this number must be specified before the generation process begins.

The final objective for this joint diffusion model is a weighted sum of the objectives for the components $L$, $F$, and $A$:

$$\mathcal{L} = \kappa_1 \mathcal{L}_L + \kappa_2 \mathcal{L}_F + \kappa_3 \mathcal{L}_A \to \min_\theta,$$

where the coefficients $\kappa_i > 0$ for $i = 1, 2, 3$ affect component prioritization, provided that we use the same neural network for making predictions.

### 2.3.2 INVARIANCES

The joint diffusion model outlined in the previous section is applied within the domain of crystals, which possesses distinct properties referred to as *symmetries*. To ensure these symmetries are preserved, the model must be parametrized in a particular manner. Further details regarding the specific types of symmetries present in the domain of crystals, as well as the organization of diffusion components required to achieve invariance to these symmetries, are provided in Appendix A.

## 3 MIRAGE INFUSION

**Motivation** In the joint diffusion model, the number of atoms in a crystal $N_{\text{atoms}}$ must be fixed in advance during the generation procedure to sample $\mathcal{M}_T$ from the prior distribution:

$$\mathcal{M}_T \sim p(\mathcal{M}_T|N_{\text{atoms}}) = p(L_T, F_T, A_T|N_{\text{atoms}}) = p(L_T)p(F_T|N_{\text{atoms}})p(A_T|N_{\text{atoms}}), \quad (1)$$

where $N_{\text{atoms}}$ is typically drawn from a categorical distribution $p(N_{\text{atoms}})$, estimated from the training data (Jiao et al., 2024a), and it usually spans a limited range of discrete values (e.g., 1 to 20). Consequently, the number of atoms in a crystal's unit cell is fixed during the whole process of generation, which, as we show in the following sections, crucially limits the model's flexibility. To address this limitation, we introduce a framework that supports adding or removing atoms during generation.

**Core idea** Traditional diffusion models cannot change object size during generation. To enable this, we reinterpret the addition and removal of atoms as transitions between different types of atoms. Specifically, we introduce *mirage atoms* – placeholder atoms that may either materialize into real atoms or vanish during the generation process.

In a crystal representation $\mathcal{M}$, mirage atoms are distinguished by assigning them a special type. Since $F$ is continuous, it cannot be used to flag mirage atoms directly. However, $A$ is discrete, so we designate a new atom type $0$ to represent a mirage atom.

**Method**  To support mirage atoms, we define an *expanded crystal domain* $\overline{\mathcal{M}} = (L, \overline{F}, \overline{A})$, where all objects have a fixed number of atoms $N_{\mathrm{m}}$ that is greater than or equal to the maximum number of atoms in a crystal in the training dataset. Mirage atoms are simply atoms with type $0$ in this domain.

We define two mappings:

- **Infusion**: Adds mirage atoms to a real crystal $\mathcal{M}$. The original atoms are kept unchanged, while the additional atoms are initialized with type $0$, and fractional coordinates are drawn from the uniform distribution over the unit cell $\mathcal{U}(0, 1)^3$.

- **Reduction**: Removes mirage atoms from expanded crystal $\overline{\mathcal{M}}$ by filtering out all atoms with type $0$, restoring the original representation $\mathcal{M}$.

Using this setup, we can train a diffusion model in the expanded domain. The model architecture remains nearly the same, with the only change being an extra atom type (type $0$) in the D3PM diffusion component for atom types.

In training, crystal from the training data $\mathcal{M}_0$ is infused with the mirage atoms with randomly initialized fractional coordinates, effectively augmenting the dataset. The lattice loss remains unchanged. The atom-type loss remains the same as well, training the model to predict real and mirage atom types in the final structure. For fractional coordinates loss, we ignore the mirage atoms defined by the mirage's mask $\mathfrak{M}_0 = \{i \mid \overline{A}_{0,i} \neq 0, \ i = \overline{1, N_{\mathrm{m}}}\}$ for the original crystal $\mathcal{M}_0$, since they have no ground-truth positions. This allows the model to freely adjust their positions and focus learning only on real atoms. The loss is masked accordingly:

---

**Algorithm 1** MiAD Training

1: **repeat**
2:    Sample $t \sim \mathcal{U}(1, T)$
3:    Sample $\mathcal{M}_0 \sim q(\mathcal{M}_0)$
4:    Infuse mirage atoms $\mathcal{M}_0 \rightarrow \overline{\mathcal{M}}_0$
5:    Add noise $\overline{\mathcal{M}}_t \sim q(\overline{\mathcal{M}}_t | \overline{\mathcal{M}}_0)$
6:    Minimize $\kappa_1 \mathcal{L}_L + \kappa_2 \mathcal{L}_{\overline{F}}^{\mathfrak{M}_0} + \kappa_3 \mathcal{L}_{\overline{A}}$
7: **until** Convergence

---

**Algorithm 2** MiAD Sampling

1: Sample $\overline{\mathcal{M}}_T \sim q(\overline{\mathcal{M}}_T)$
2: **for** $t \leftarrow T$ **to** $1$ **do**
3:    Denoise $\overline{\mathcal{M}}_{t-1} = q(\overline{\mathcal{M}}_{t-1} | \overline{\mathcal{M}}_t)$
4: **end for**
5: Reduce $\overline{\mathcal{M}}_0 \rightarrow \mathcal{M}_0$

---

$$\mathcal{L}_{\overline{F}}^{\mathfrak{M}_0} = \mathbb{E}_{\overline{\mathcal{M}}_0 \sim q(\overline{\mathcal{M}}_0), t \sim \mathcal{U}(1,T), \overline{\mathcal{M}}_t \sim q(\overline{\mathcal{M}}_t | \overline{\mathcal{M}}_0)} \sum_{i \in \mathfrak{M}_0} ||\nabla_{\overline{F}_{t,i}} \log q(\overline{F}_t | \overline{F}_0) - s_{\theta,i}(\overline{\mathcal{M}}_t, t)||_2^2, \quad (2)$$

where $q(\overline{\mathcal{M}}_0)$ is a distribution of crystals after the infusion of mirage atoms, and $s_{\theta,i}(\overline{\mathcal{M}}_t, t)$ is a score for the fractional coordinates of atom $i$.

The sampling algorithm remains the same, except for two differences: (1) at the start of generation, all crystals are sampled with the same number of atoms $N_{\mathrm{m}}$ (not with $N_{\mathrm{atoms}} \sim p(N_{\mathrm{atoms}})$ as in 1), (2) at the end of generation we need to apply the reduction operator to project the crystal onto the original domain.

Importantly, this method preserves existing symmetries because lattice representations remain unchanged and mirage atoms follow the same spatial rules as real atoms.

We call the proposed technique *mirage infusion*. It expands the space of generative trajectories available to the model, increasing its flexibility and expressiveness. We outline the training procedure in Algorithms 1, and the sampling procedure in Algorithm 2, as well as their schematic visualization in Figure 1.

**Discussion**  There are several design choices when defining the expanded domain. In our setup, we: (1) initialize mirage atoms with uniformly random coordinates. This choice ensures that mirage atoms can occupy arbitrary positions in the unit cell and introduces variability into the generation process. (2) Mask the loss for mirage atoms during training to avoid learning from noise. This allows the model to focus on denoising atoms that exist in the final structure while giving it the freedom to learn when and how mirage atoms should transform into real ones during generation. Skipping the mask would force the model to predict a zero score for mirage atoms, potentially reducing expressiveness.

A similar idea was proposed independently by Schneuing et al. (2025) for structure-based drug design. In their method, mirage atom coordinates are initialized at the object's center of mass, and no loss masking is applied. While both methods share conceptual goals, we argue that our design is more principled and flexible. Empirical comparisons support this claim, showing that our approach achieves higher generative quality (see Appendix B.3).

## 4 RELATED WORK

The field of deep generative modeling in material science has been rapidly growing in recent years. State-of-the-art approaches use a multi-step procedure for generating crystals. In this review, we organize the works by their treatment of the number of atoms during this generation process.

Models with a constant number of atoms directly sampled from the prior distribution; their types and coordinates are gradually refined during the generation process along with the lattice. DiffCSP (Jiao et al., 2024a) is the first pure diffusion model for crystal generation, our work directly extends their approach. Most parts of this model are described in Section 2; briefly, it is a joint diffusion model that operates with the crystal space represented as a triplet of lattice, fractional coordinates, and atom types. The following models fall within the same paradigm: MatterGen (Zeni et al., 2024), FlowMM (Miller et al., 2024), CrysBFN (Wu et al., 2025), TGDMat (Das et al., 2025), CrystalFlow (Luo et al., 2025). DiffCSP++ (Jiao et al., 2024b) is a development of DiffCSP that constrains the diffusion process so that the crystal has predefined Wyckoff symmetries, for de novo generation they are sampled from the training dataset. ADiT (Joshi et al., 2025), uses a Transformer instead of a GNN as the denoising model.

Models with two stages, where the first stage is used to generate an intermediate crystal representation that includes the number of atoms, from which the structure is reconstructed during the second stage, where the number of atoms is fixed: CDVAE (Xie et al., 2022), FlowLLM (Sriram et al., 2024), WyFormer (Kazeev et al., 2025). All these models combine non-diffusion generative models in the first stage with diffusion-based refinement in the second stage.

Models that change the number of atoms during the entire generation process, i.e., the number of atoms is generated jointly with all other components of the crystal.

The first subgroup consists of methods for changing the representation of 3D crystal structures, which allows them to vary the number of atoms during the generation process, Uni-3DAR (Lu et al., 2025), UniMat (Yang et al., 2024), WyckoffDiff (Kelvinius et al., 2025). Autoregressive models naturally fall into this subgroup: CrystalFormer (Cao et al., 2024), CrystaLLM (Antunes et al., 2024), LLaMA-2 (Gruver et al., 2024).

The second subgroup consists of Crystal-GFN (AI4Science et al., 2023) and SHAFT (Nguyen et al., 2024) – policies, which are trained using reward functions and operate in a space of 3D crystal representations along with their symmetry groups.

The third subgroup consists of models, which indirectly change the number of atoms during generation via changing Wyckoff positions site symmetry: SymmCD (Levy et al., 2025) and SymmBFN (Ruple et al., 2025). The diffusion/flow process, however, only deals with an asymmetric unit of fixed size.

The proposed categorization underscores the inherent limitations of diffusion-like methods in accommodating the insertion and removal of atoms during the generation process. This finding is particularly noteworthy, as leading approaches such as DiffCSP, FlowLLM, WyFormer, and ADiT are either partially or entirely based on the diffusion paradigm.

## 5 METRICS

De novo crystal generation serves as the first step of the material discovery pipeline. Xie et al. (2022) proposed several metrics for evaluating generative models in this task: Structural and Compositional Validity, Coverage-Recall, Coverage-Precision, and Wasserstein distances between the distributions of various properties in the training set and those of crystals produced by the model. Although these metrics have become widespread tools for model comparison, they have severe limitations that constrain their applicability to analyzing the performance of modern generative models. We report MiAD's results under these metrics, as well as our arguments against their use in their current

form, in Appendix E. Among existing approaches for comparing generative models in de novo crystal generation, we focus on S.U.N. (Zeni et al., 2024) — the proportion of materials that are simultaneously *stable*, *unique* and *novel*. This is one of the most reasonable measures of a generative model's performance, as it directly quantifies the model's utility for materials discovery.

**Novelty** Represents the fraction of the generated materials that are not present in the training dataset. Following all the baselines, we use `StructureMatcher` from the `pymatgen` package (Ong et al., 2013) with the default parameters.

**Uniqueness** As the number of generated materials grows, a model starts to repeat itself. Uniqueness is the fraction of unique materials among generated, also estimated with `StructureMatcher`.

**Stability** To be useful, a material must actually exist under normal conditions. Ab initio prediction of experimental stability is an open research question (Tolborg et al., 2022; Sun et al., 2016) with various trade-offs between the accuracy and computational cost possible. Again, we follow state-of-the-art ML baselines (Miller et al., 2024; Kazeev et al., 2025; Zeni et al., 2024; Joshi et al., 2025) and use *energy above convex hull $E^{\text{hull}}$* as the stability measure. There are two important nuances.

(1) Stability condition If a material has positive $E^{\text{hull}}$, this indicates that the same set of atoms can be rearranged into a known different configuration with a lower potential energy. This, however, does not necessarily mean that the higher-energy material will not exist under normal conditions; for example, both graphite and diamond do. We, therefore, measure the number of both *stable* materials with $E^{\text{hull}} < 0$ eV that are highly likely to exist; and *metastable* materials with $E^{\text{hull}} < 0.1$ eV, which are likely, but not certain to exist; the choice of the threshold follows Joshi et al. (2025).

(2) Energy computation method Density Functional Theory (DFT) (Kohn & Sham, 1965) is the ab initio method that has been used to obtain the energy values and structures in Materials Project (Jain et al., 2013b), used both as training data and convex hull. We use DFT to support the main claims of our paper, to evaluate the performance of MiAD, and compare it to the baselines in Table 1; the computation details are described in Appendix D. Due to the large computational cost of DFT, not all baseline works use it for stability evaluation, some opt for Machine Learning Interatomic Potentials (MLIPs). MLIPs are a rapidly advancing research area, useful for stability estimation (Riebesell et al., 2023), but still fall short of DFT accuracy (Deng et al., 2025), as evident by S.U.N. values difference for the same models in Tables 1 and 2. We use two MLIP models, CHGNet (Deng et al., 2023) and eq-V2 (Barroso-Luque et al., 2024), for ablation studies and for a supplementary comparison with baseline methods for which DFT data are not available. In the cases of DFT or CHGNet, we first prerelax crystals via CHGNet for 1500 steps, while in the cases of eq-V2, we prerelax via eq-V2 for 100 steps.

# 6 EXPERIMENTS

In our experimental evaluation, we aim to highlight the importance of the model's capability to modify the number of atoms during the generative process. The proposed approach is a combination of the DiffCSP model Jiao et al. (2024a) with the proposed mirage infusion technique (see Section 3). To isolate the effect of this technique, we retain an identical neural network architecture, thereby minimizing confounding variables.

However, it should be noted that the proposed model incurs higher computational costs, attributable to the increased average number of atoms per generated crystal. Additionally, mirage infusion necessitates a greater number of optimization iterations to attain optimal performance. All experiments are conducted using the MP-20 dataset (Jain et al., 2013a), and results are

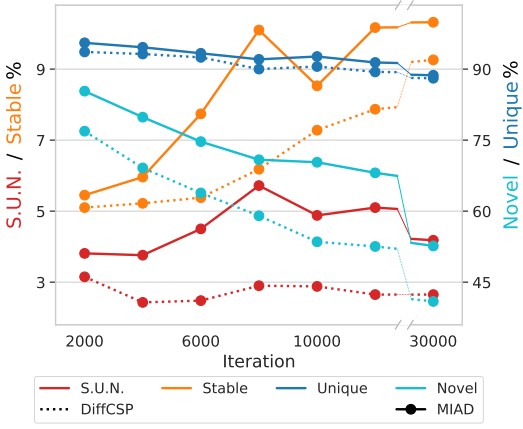

Figure 2: Comparison of MiAD (DiffCSP with mirage infusion) and DiffCSP in terms of stability, uniqueness, novelty, and S.U.N. Stability is estimated via eq-V2.

Table 1: **Crystal generation comparison via S.U.N. based on DFT.** We report metastability, M.S.U.N., stability, and S.U.N. for $10\,000$ sampled crystals. For clarity in evaluating the model's quality, we also report the Unique&Novel rate separately for metastable and stable crystals, respectively. MiAD outperforms all existing approaches in terms of M.S.U.N. and S.U.N. DiffCSP & FlowMM results are taken from Miller et al. (2024), FlowLLM Sriram et al. (2024), ADiT Joshi et al. (2025); WyFormer computed by us from the DFT structures provided by the authors.

| Model | Metastability ($E^{\text{hull}} < 0.1$) | | | Stability ($E^{\text{hull}} < 0.0$) | | |
|---|---|---|---|---|---|---|
| | Metastable (%) ↑ | Unique&Novel (%) ↑ | M.S.U.N. (%) ↑ | Stable (%) ↑ | Unique&Novel (%) ↑ | S.U.N. (%) ↑ |
| DiffCSP | - | - | - | 5.0 | 66.0 | 3.3 |
| FlowMM | 30.6 | 73.5 | 22.5 | 4.6 | 60.9 | 2.8 |
| FlowLLM | 66.9 | 39.3 | 26.3 | 13.9 | 33.8 | 4.7 |
| WyFormer | 30.5 | **89.8** | 27.4 | 5.2 | **92.3** | 4.8 |
| MP20-only ADiT | **81.6** | 31.8 | 25.9 | 14.1 | 33.3 | 4.7 |
| MP20-only ADiT (32M) | 71.1 | 53.6 | 38.1 | 12.8 | 50.8 | 6.5 |
| Jointly trained ADiT | 81.0 | 34.8 | 28.2 | **15.4** | 34.4 | 5.3 |
| MiAD | 73.5 | 59.4 | **43.6** | 12.5 | 65.2 | **8.2** |

Table 2: **Crystal generation comparison via S.U.N. based on MLIPs.** We report stability and S.U.N. estimated using CHGNet and eq-V2 for $10\,000$ sampled crystals. For clarity in evaluating the model's quality, we also report the Unique&Novel rate among stable crystals. MiAD outperforms all existing approaches in terms of both variants of S.U.N. Results for eq-V2 are computed by us from the structures provided by the authors, whereas results for CHGNet are taken from Levy et al. (2025) and Ruple et al. (2025).

| Model | eq-V2 ($E^{\text{hull}} < 0.0$) | | | CHGNet ($E^{\text{hull}} < 0.0$) | | |
|---|---|---|---|---|---|---|
| | Stable (%) ↑ | Unique&Novel (%) ↑ | S.U.N. (%) ↑ | Stable (%) ↑ | Unique&Novel (%) ↑ | S.U.N. (%) ↑ |
| CDVAE | - | - | - | 4.4 | **96.4** | 4.3 |
| DiffCSP | 3.8 | 66.6 | 2.5 | 11.3 | 78.8 | 8.9 |
| DiffCSP++ | 2.9 | 73.5 | 2.1 | 11.4 | 75.9 | 8.6 |
| FlowMM | 2.0 | 70.4 | 1.4 | 9.1 | 71.7 | 6.5 |
| SymmCD | 2.8 | 70.3 | 2.0 | 9.3 | 73.5 | 6.9 |
| SymmBFN | - | - | - | 11.8 | 75.4 | 8.9 |
| MatterGen MP-20 | 2.5 | **73.6** | 1.8 | 9.7 | 83.5 | 8.1 |
| MiAD | **9.7** | 57.0 | **5.5** | 19.8 | 65.2 | **12.9** |

assessed using various versions of the S.U.N. metric (see Section 5). Other experimental details are provided in Appendix C.

Appendix B provides a detailed exposition of key design choices underlying the mirage infusion technique. Specifically, we illustrate that, within the proposed definition of the expanded domain, there exists flexibility in selecting $N_{\text{m}}$, the total number of real and mirage atoms for crystals in the expanded domain. Moreover, mirage infusion influences the scaling of loss terms associated with fractional coordinates and atom types. Our analysis reveals that suboptimal weighting of these loss components in the joint diffusion model can have a nontrivial impact on overall model quality. As discussed in Section 4, alternative formulations of mirage infusion have been proposed. We present a comparison between our definition and a contemporary variant, underscoring the critical role of expanded domain design and loss function modifications. Lastly, we compare the final version of mirage infusion with the baseline model without these modifications, thereby demonstrating the substantial benefits of the proposed approach simultaneously in stability, uniqueness, and novelty rate (see Figure 3). This ablation is performed using S.U.N., where stability is estimated using eq-V2 (Barroso-Luque et al., 2024) due to computational constraints.

We designate the finalized model as MiAD (Mirage Atom Diffusion) and benchmark its performance against state-of-the-art methods for de novo crystal generation. Multiple versions of the S.U.N. metric are employed to facilitate comprehensive comparisons across a broad range of existing models. It should be noted that we do not compare against studies that lack S.U.N. metric evaluations.

Comparison in Table 1 presents the results, demonstrating MiAD performance against existing approaches, with respect to the S.U.N. metric, where stability is assessed via DFT (Kohn & Sham, 1965) according to the protocol described in Appendix D. As outlined in Section 5, this configuration represents the most rigorous S.U.N. evaluation and currently prevails over all other approaches. While MiAD exhibits a lower fraction of stable crystals compared to ADiT (Joshi et al., 2025), this is

attributable to the tendency of this model to replicate training set samples. Consequently, stability alone is insufficient as a measure of generative model quality. At the same time, MiAD exhibits a lower fraction of unique and novel crystals among stable compared to WyFormer, while exceeding the last one in terms of stability rate. Therefore, MiAD achieves superior overall S.U.N. performance relative to all baselines ($+25\%$ relative to the closest competing method) due to the best trade-off between stability, uniqueness, and novelty. Another significant observation is that the incorporation of mirage infusion enhances the performance of the original DiffCSP model by up to $\times 2.5$ times, representing a substantial improvement.

Furthermore, some works only report S.U.N. computed with MLIPs, presumably due to computational constraints. MiAD comparison to them is presented in Table 2; that MiAD shows the best performance there as well.

Finally, we provide additional analyses of the diversity of MiAD's generations. In Appendix F, we demonstrate that MiAD changes the number of atoms during generation and successfully produces S.U.N. crystals with varying numbers of atoms. In Appendix G, we provide the distributions of space groups for modern generative models and verify that MiAD preserves diversity in these terms. The scalability of the proposed model on larger datasets is demonstrated in Appendix H, where we compare MiAD with MatterGen on the Alex-MP20 dataset (Zeni et al., 2024).

## 7 DISCUSSION

**Conclusion**  The categorization provided in Section 4 offers valuable insights into the potential applicability of the proposed technique across a range of models within the domain of de novo material generation. Beyond DiffCSP, which serves as the foundation for the model introduced in this work, the mirage infusion technique can be directly implemented, without modification, in MatterGen, FlowMM, and CrystalFlow, given that these models employ an identical crystal representation and share the same factorization of the loss function. Additionally, we hypothesize that an adapted version of this technique could be extended to models such as ADiT, CDVAE, FlowLLM, SymmCD, and SymmBFN, although further investigation is required to assess its effectiveness, and the eventual impact in these cases remains uncertain. At the same time, in Section 6 we present MiAD, demonstrating that mirage infusion gives a substantial boost in quality for the base joint diffusion model and outperforms existing state-of-the-art approaches. This improvement demonstrates the potential of approaches directed at the modification of the space with which a particular diffusion model works.

**Limitations**  The proposed technique, mirage infusion, demonstrates a substantial improvement in the specific joint diffusion model for crystal generation, as evidenced by the S.U.N. metric. Nevertheless, further research is essential to investigate various adaptations of this technique, particularly its application to other established generative models in the domain of de novo crystal generation, as well as in other areas of generative modeling, to more comprehensively assess the methodology's position within the broader field. Additionally, the field of de novo crystal generation requires the development of new metrics capable of evaluating the diversity of existing generative models from a wider range of perspectives, thereby enhancing the understanding of the properties of both existing and newly generative models. At present, the most effective available metric is employed to assess the impact of the proposed technique, with a primary focus on illustrating its effects in this particular context.

**Societal impact**  Mirage Atom Diffusion advances de novo crystalline material generation by letting diffusion models vary atom counts during synthesis, boosting diversity and quality and accelerating discoveries for clean energy, electronics, and medicine. Its compatibility with multiple generative frameworks broadens access. But greater flexibility brings risks: potential design of hazardous or harmful materials without strong oversight, bias from narrow metrics like S.U.N., and high environmental and financial costs to validate many candidates. Growing model complexity also heightens concerns over interpretability, reproducibility, and equitable compute access.

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

# A CRYSTAL INVARIANCES

## A.1 SYMMETRIES OF CRYSTAL STRUCTURE DISTRIBUTION

The space of 3D crystal structures is governed by symmetries that impose stringent constraints and play a crucial role in directing the generation process from the distribution $p(\mathcal{M})$. The model, proposed in Jiao et al. (2024a) and described in Section 2.3, focuses on the following symmetries:

**Permutation invariance:** $\forall$ permutation $P \in S_N \implies p(L, FP, AP) = p(L, F, A)$

**$O(3)$ invariance:** $\forall$ orthogonal transformation $Q \in \mathbb{R}^{3 \times 3} \implies p(QL, F, A) = p(L, F, A)$

**Periodic translation invariance:** $\forall$ translation $\tau \in \mathbb{R}^{1 \times 3} \implies p(L, w(F + \mathbf{1}\tau), A) = p(L, F, A)$

Where $w(F) = F - \lfloor F \rfloor \in [0, 1)^{N_{\text{atoms}} \times 3}$ returns the fractional part of each element in $F$ and $\mathbf{1} \in \mathbb{R}^{N_{\text{atoms}} \times 1}$ is a unit vector.

Informally, each symmetry signifies that performing a particular transformation on a crystal does not alter its likelihood under the distribution $p(\mathcal{M})$. These properties provide a strong inductive bias and can be methodically integrated into the diffusion model by design.

## A.2 INVARIANT DIFFUSION MODELS

A diffusion model $p_\theta(x_0)$ is said to be invariant with respect to the symmetry group $\mathbb{G}$ if, for any transformation $g \in \mathbb{G}$, it holds that $p_\theta(g \cdot x_0) = p_\theta(x_0)$. Hoogeboom et al. (2022) proposed that invariance can be encapsulated in the diffusion model if we define:

$$\textbf{Invariant prior distribution:} \ \forall \ g \in \mathbb{G} \implies p(x_T) = p(g \cdot x_T) \tag{3}$$

$$\textbf{Equivariant transition kernels:} \ \forall \ g \in \mathbb{G} \implies p_\theta(g \cdot x_{t-1} | g \cdot x_t) = p_\theta(x_{t-1} | x_t) \tag{4}$$

$$q(g \cdot x_t | g \cdot x_{t-1}) = q(x_t | x_{t-1}) \tag{5}$$

If conditions 3, 4 hold, then the diffusion model $p_\theta(x_0)$ is invariant with respect to the symmetry group $\mathbb{G}$:

$$p_\theta(g \cdot x_0) = \int p(g \cdot x_T) \prod_{t=1}^{T} p_\theta(g \cdot x_{t-1} | g \cdot x_t) \, dx_{1:T} = \int p(x_T) \prod_{t=1}^{T} p_\theta(x_{t-1} | x_t) \, dx_{1:T} = p_\theta(x_0),$$

whereas if conditions 4, 5 hold, then the conditional backward process is equivariant:

$$q(g \cdot x_{t-1} | g \cdot x_t, g \cdot x_0) = \frac{q(g \cdot x_t | g \cdot x_{t-1}) q(g \cdot x_{t-1} | g \cdot x_0)}{q(g \cdot x_t | g \cdot x_0)} =$$

$$= \frac{q(x_t | x_{t-1}) q(x_{t-1} | x_0)}{q(x_t | x_0)} = q(x_{t-1} | x_t, x_0),$$

and the training objective is invariant:

$$\mathcal{L}_g = \sum_{t=2}^{T} \gamma_t \mathbb{E}_{g \cdot x_0 \sim q(g \cdot x_0), g \cdot x_t \sim q(g \cdot x_t | g \cdot x_0)} D_{\text{KL}} \left[ q(g \cdot x_{t-1} | g \cdot x_t, g \cdot x_0) \, || \, p_\theta(g \cdot x_{t-1} | g \cdot x_t) \right] =$$

$$= \sum_{t=2}^{T} \gamma_t \mathbb{E}_{x_0 \sim q(x_0), x_t \sim q(x_t | x_0)} D_{\text{KL}} \left[ q(x_{t-1} | x_t, x_0) \, || \, p_\theta(x_{t-1} | x_t) \right] = \mathcal{L}$$

The same remains true for score-matching objective if in addition to the condition 5 the following conditions are also satisfied: 6 – equivariance of the score-estimator $s_\theta(x_t, t)$, 7 – distributivity of the transformations $g$ from the group $\mathbb{G}$, 8 – invariance of $l_2$-norm:

$$s_\theta(g \cdot x_t, t) = g \cdot s_\theta(x_t, t) \tag{6}$$

$$g \cdot x + g \cdot y = g \cdot (x + y) \tag{7}$$

$$||x||_2^2 = ||g \cdot x||_2^2 \tag{8}$$

Then the score-matching objective is invariant with respect to the symmetry group $\mathbb{G}$:

$$\mathcal{L}_g = \mathbb{E}_{g \cdot x_0 \sim q(g \cdot x_0), t \sim \mathcal{U}(1,T), g \cdot x_t \sim q(g \cdot x_t | g \cdot x_0)} || \nabla_{g \cdot x_t} \log p(g \cdot x_t | g \cdot x_0) - s_\theta(g \cdot x_t, t) ||_2^2 =$$

$$= \mathbb{E}_{x_0 \sim q(x_0), t \sim \mathcal{U}(1,T), x_t \sim q(x_t | x_0)} || g \cdot \nabla_{x_t} \log p(x_t | x_0) - g \cdot s_\theta(x_t, t) ||_2^2 =$$

$$= \mathbb{E}_{x_0 \sim q(x_0), t \sim \mathcal{U}(1,T), x_t \sim q(x_t | x_0)} || g \cdot (\nabla_{x_t} \log p(x_t | x_0) - s_\theta(x_t, t)) ||_2^2 =$$

$$= \mathbb{E}_{x_0 \sim q(x_0), t \sim \mathcal{U}(1,T), x_t \sim q(x_t | x_0)} || \nabla_{x_t} \log p(x_t | x_0) - s_\theta(x_t, t) ||_2^2 = \mathcal{L}$$

The definition of the diffusion model in terms of the score function allows for the formulation of various backward transition kernels, denoted as $K_\theta$:

$$x_{t-1} = K_\theta(x_t)$$

These kernels incorporate the score estimator $s_\theta$ in a specific manner and can be either deterministic, or stochastic. Equivariance with respect to the symmetry group $\mathbb{G}$, in the context of transition kernels $K$, implies the following condition in the deterministic case:

$$\forall g \in \mathbb{G}, \quad K(g \cdot x) = g \cdot K(x) \tag{9}$$

In the stochastic case, this equality is sufficient. However, the necessary and sufficient condition requires equality solely in terms of probability density functions.

### A.2.1 INVARIANCES

Jiao et al. (2024a) proposed that symmetric properties outlined in Section A.1 can be encapsulated in the joint diffusion model described in Section 2.3 via specific neural network parameterization.

**Permutation invariance** affects the components $F$ and $A$. The prior distributions for these components are defined element-wise, and the corresponding transition kernels remain equivariant as long as the neural network is equivariant. It is achieved using a graph neural network (GNN) architecture that alternates between message-passing and atom-wise processing layers. The message-passing layer operates on atom pairs selected based on rules that are independent of their position in the sequence, i.e., distance cutoff between atoms. As a result, reordering of the input sequence of atoms results in the same reordering of the output sequence of predictions, ensuring permutation equivariance.

**$O(3)$ invariance** affects only the $L$ component of the crystal because the $F$ component defines coordinates within the unit cell and does not contain any information about the orientation of the unit cell in space. The prior distribution in DDPM is already invariant. In order to define an equivariant transition kernel we need to parametrize its mean in the following way:

$$\mu_\theta(\mathcal{M}_t, t) = \mu_\theta(L_t, F_t, A_t, t) = L_t \text{NN}_\theta(L_t^T L_t, F_t, A_t, t),$$

where $\text{NN}_\theta : \mathcal{M}_t, t \to \mathbb{R}^{3 \times 3}$ – neural network with a linear layer on the top. Then, the following transition kernel in DDPM:

$$p_\theta(L_{t-1} | \mathcal{M}_t) = p_\theta(L_{t-1} | L_t, F_t, A_t) = \mathcal{N}(L_{t-1} | \mu_\theta(L_t, F_t, A_t, t), \sigma_t I)$$

is $O(3)$ equivariant:

$$p_\theta(QL_{t-1} | QL_t, F_t, A_t) = \mathcal{N}(QL_{t-1} | \mu_\theta(QL_t, F_t, A_t, t), \sigma_t I) =$$

$$= \mathcal{N}(QL_{t-1} | QL_t \text{NN}_\theta(L_t^T Q^T QL_t, F_t, A_t, t), \sigma_t I) =$$

$$= \mathcal{N}(QL_{t-1} | QL_t \text{NN}_\theta(L_t^T L_t, F_t, A_t, t), \sigma_t I) =$$

$$= \mathcal{N}(QL_{t-1} | Q\mu_\theta(L_t, F_t, A_t, t), \sigma_t I) =$$

$$= \mathcal{N}(L_{t-1} | \mu_\theta(L_t, F_t, A_t, t), \sigma_t I) =$$

$$= p_\theta(L_{t-1} | L_t, F_t, A_t)$$

**Periodic translation invariance** affects only the $F$ component. The prior distribution in Wrapped Normal diffusion is already invariant. To ensure that the transition kernel is equivariant, we parameterize the score estimator $s_\theta$ in an invariant form:

$$s_\theta(\mathcal{M}_t, t) = s_\theta(L_t, F_t, A_t, t) = s_\theta\Big(L_t, \text{PairwiseDist}(F_t), A_t, t\Big),$$

$$\text{PairwiseDist}(F_t) = \big\{ \psi_{FT}(f_j - f_i) \,\big|\, i, j = \overline{1, N_{\text{atoms}}}, i \neq j \big\},$$

where $f_i$ – fractional coordinates of the atom $i$ from the $F_t$, $\psi_{FT} : (-1,1)^3 \rightarrow [-1,1]^{3 \times K}$ – the Fourier Transformation of the relative fractional coordinate $f_i - f_j$, and PairwiseDist : $[0,1]^{N_{\text{atoms}} \times 3} \rightarrow [-1,1]^{N_{\text{atoms}} \times N_{\text{atoms}} \times K}$ – yields coordinates representation which is invariant to periodic translations:

$$\text{PairwiseDist}\left(w\left(F_t + \mathbf{1}\tau\right)\right) = \text{PairwiseDist}(F_t)$$

This follows from the fact that pairwise atomic distances remain unchanged when the entire system of atoms is translated. Further, if we use the following form of the stochastic backward transition kernel $K_\theta$:

$$F_{t-1} = K_\theta(\mathcal{M}_t) = K_\theta(L_t, F_t, A_t) = w\left(a_t F_t + b_t s_\theta(L_t, F_t, A_t, t) + c_t \epsilon\right), \quad \epsilon \sim \mathcal{N}(0,1),$$

where $a_t, b_t, c_t$ are a scalar coefficients, then $K_\theta$ is periodic translation equivariant (sufficient condition 9):

$$
\begin{aligned}
K_\theta\left(L_t, w\left(F_t + \mathbf{1}\tau\right), A_t\right) &= w\left(a_t w\left(F_t + \mathbf{1}\tau\right) + b_t s_\theta(L_t, w\left(F_t + \mathbf{1}\tau\right), A_t, t) + c_t \epsilon\right) = \\
&= w\left(a_t w\left(F_t + \mathbf{1}\tau\right) + b_t s_\theta(L_t, F_t, A_t, t) + c_t \epsilon\right) = \\
&= w\left(a_t F_t + b_t s_\theta(L_t, F_t, A_t, t) + c_t \epsilon + \mathbf{1}\tau\right) = \\
&= w\left(w\left(a_t F_t + b_t s_\theta(L_t, F_t, A_t, t) + c_t \epsilon\right) + \mathbf{1}\tau\right) = \\
&= w\left(K_\theta(L_t, F_t, A_t) + \mathbf{1}\tau\right)
\end{aligned}
$$

# B  ABLATION

As discussed in Section 6, several critical steps were undertaken in developing the final version of the MiAD.

## B.1  NUMBER OF MIRAGE ATOMS

The definition of mirage infusion necessitates setting the hyperparameter $N_{\text{m}}$, which specifies that crystals are supplemented with mirage atoms until the total number of atoms reaches $N_{\text{m}}$. Increasing this hyperparameter results in a greater number of possible atom variants from which the model can select during crystal construction. Simultaneously, it increases the number of atoms the model must eliminate during the generation process. The neural network architecture proposed by Jiao et al. (2024a) has a limit on the number of atoms that can interact efficiently. Beyond a certain number, the neural network loses the capacity to manage them effectively. We evaluate several variants of $N_{\text{m}}$ in Table 3 using S.U.N. computed via MLIPs (see Section 5), and select the optimal variant for all subsequent experiments.

## B.2  LOSS COMPONENT PRIORITIZATION

The application of mirage infusion alters the loss components associated with diffusion in fractional coordinates and atom types. This, in turn, affects the scale of the gradients and the prioritization of tasks the neural network must solve concurrently: 1) lattice prediction, 2) fractional coordinates prediction, and 3) atom types prediction. We discovered that the balance among these tasks, especially the influence of the loss component associated with atom types $\mathcal{L}_A$ (see Section 2.3.1), can

Table 3: **Ablation study on the number of mirage atoms in MiAD** We perform a comparison of various values for $N_{\text{m}}$ within MiAD: $20, 25, 30, 35$. This hyperparameter specifies that crystals are supplemented with mirage atoms until the total number of atoms reaches $N_{\text{m}}$. The models are compared using S.U.N. for $10\,000$ sampled crystals, where stability is estimated via (left) eq-V2 and (right) CHGNet.

| | eq-V2 ($E^{\text{hull}} < 0.0$) | | | | CHGNet ($E^{\text{hull}} < 0.0$) | | | |
|---|---|---|---|---|---|---|---|---|
| Model | Stable (%) ↑ | Unique (%) ↑ | Novel (%) ↑ | S.U.N. (%) ↑ | Stable (%) ↑ | Unique (%) ↑ | Novel (%) ↑ | S.U.N. (%) ↑ |
| MiAD (20) | 8.7 | 92.0 | 73.0 | 5.3 | 17.4 | 91.9 | 72.9 | 12.0 |
| **MiAD (25)** | **9.7** | 92.2 | 71.1 | **5.5** | **19.8** | 92.2 | 71.3 | **12.9** |
| MiAD (30) | 8.3 | **93.1** | **74.1** | 4.7 | 16.6 | **93.2** | **73.9** | 11.3 |
| MiAD (35) | 8.2 | 92.3 | 70.9 | 4.6 | 17.9 | 92.4 | 70.9 | 11.8 |

Table 4: **Ablation study of loss scaling for atom types in MiAD** We perform a comparison of the coefficients $0.5, 1.0, 2.0$ applied to the loss function for atom types in MiAD, while maintaining constant scales for the losses associated with lattice and fractional coordinates. We quantify the prioritization of the loss components corresponding to $(L - F - A)$ at the end of the training procedure as a percentage of the total loss. The prioritization for these coefficients are as follows: $\mathcal{L}_A \times 0.5$: $(40 - 50 - 10)$, $\mathcal{L}_A \times 1.0$: $(36 - 46 - 18)$, $\mathcal{L}_A \times 2.0$: $(31 - 39 - 30)$. The models are compared using S.U.N. for $10\,000$ sampled crystals, where stability is estimated via (left) eq-V2 and (right) CHGNet.

| | eq-V2 ($E^{hull} < 0.0$) | | | | CHGNet ($E^{hull} < 0.0$) | | | |
| Model | Stable (%) ↑ | Unique (%) ↑ | Novel (%) ↑ | S.U.N. (%) ↑ | Stable (%) ↑ | Unique (%) ↑ | Novel (%) ↑ | S.U.N. (%) ↑ |
|---|---|---|---|---|---|---|---|---|
| MiAD ($\mathcal{L}_A \times 0.5$) | 8.2 | **93.8** | **77.6** | 4.7 | 17.9 | **93.8** | **77.6** | 12.3 |
| **MiAD ($\mathcal{L}_A \times 1.0$)** | **9.7** | 92.2 | 71.1 | **5.5** | **19.8** | 92.2 | 71.3 | **12.9** |
| MiAD ($\mathcal{L}_A \times 2.0$) | 8.9 | 91.5 | 67.4 | 5.0 | 19.0 | 91.5 | 67.4 | 11.9 |

Table 5: **Ablation study of possible definitions of the mirage infusion** We perform a comparison of the definitions of MiAD in terms of (1) the initialization of fractional coordinates of mirage atoms in crystals: sampling from the uniform distribution or positioning at the geometric center of mass of the real atoms, (2) the masking of mirage atoms in the loss for fractional coordinates. The models are compared using S.U.N. for $10\,000$ sampled crystals, where stability is estimated via (left) eq-V2 and (right) CHGNet.

| | eq-V2 ($E^{hull} < 0.0$) | | | | CHGNet ($E^{hull} < 0.0$) | | | |
| Model | Stable (%) ↑ | Unique (%) ↑ | Novel (%) ↑ | S.U.N. (%) ↑ | Stable (%) ↑ | Unique (%) ↑ | Novel (%) ↑ | S.U.N. (%) ↑ |
|---|---|---|---|---|---|---|---|---|
| **MiAD (Uniform + Masked)** | **9.7** | 92.2 | 71.1 | **5.5** | **19.8** | 92.2 | 71.3 | **12.9** |
| MiAD (Uniform + NonMasked) | 7.7 | 91.7 | 65.1 | 3.8 | 16.6 | 91.6 | 65.2 | 10.3 |
| MiAD (Center + Masked) | 0.5 | 93.4 | **93.1** | 0.4 | 5.8 | 92.9 | **93.0** | 5.2 |
| MiAD (Center + NonMasked) | 2.6 | **96.3** | 87.9 | 1.3 | 9.9 | **96.3** | 87.9 | 8.1 |

substantially affect the model's quality. During our experiments, we did not modify the scales of the loss components related to lattice and fractional coordinates in order to clearly demonstrate the impact of the proposed technique. However, the loss associated with atom types is significantly influenced by the hyperparameter $N_m$ in mirage infusion, which raises the question of whether additional corrections to this component are necessary. We quantify the prioritization of the loss components corresponding to $(L - F - A)$ at the end of the training procedure as a percentage of the total loss. The application of mirage infusion with $N_m = 25$ reduces by half the loss for atom types, denoted as $\mathcal{L}_A$, and leads to a balance of $(36 - 46 - 18)$ among MiAD components, whereas the balance among components in the original DiffCSP is $(31 - 39 - 30)$. Then, in Table 4, we compare MiAD models, where $\mathcal{L}_A$ is further increased or decreased by a factor of two, however, it only diminishes the quality. We considered it essential to illustrate the effects of these adjustments, due to the scale of an impact on the model's quality. In Table 4, the impact of $\mathcal{L}_A$ as a percentage of the total loss is not precisely doubled or halved because alterations to this component also influence the total loss. Given the significance of this loss prioritization, we conducted experiments, the results of which are presented in Table 3, using coefficients for $\mathcal{L}_A$ that maintain the same balance of $(36 - 46 - 18)$ among the loss components by the end of the training procedure.

## B.3 DEFINITIONS OF MIRAGE INFUSION

As discussed in Section 3, Schneuing et al. (2025) introduced a related concept involving the augmentation of original molecular structures with hypothetical (non-existent) components for structure-based drug design. In this approach, the fractional coordinates of mirage atoms are initialized at the geometric center of mass of the real atoms, and no masking is applied to the mirage atoms. Table 5 compares this formulation with the definition of mirage infusion proposed in the present work, showing that the latter achieves substantially improved performance.

## B.4 FINAL IMPACT OF MIRAGE INFUSION

Figure 3 illustrates the comparison between the final version of MiAD and the original DiffCSP. In this analysis, we employed the optimal variant of $N_m = 25$ as identified in Table 3 and adhered to the

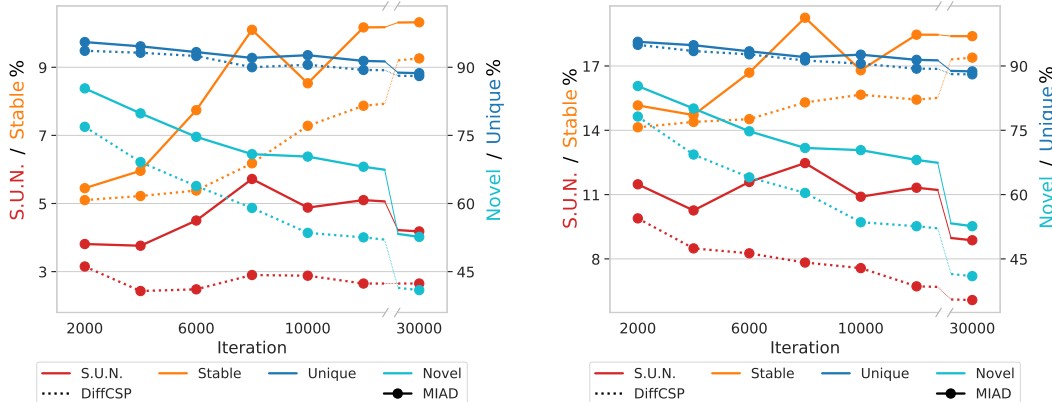

Figure 3: Comparison of MiAD (DiffCSP with mirage infusion) in its final version and DiffCSP in terms of stability, uniqueness, novelty, and S.U.N. Stability is estimated via (left) eq-V2 and (right) CHGNet. MiAD outperforms DiffCSP across all metrics, especially in terms of stability rate and S.U.N., achieving the highest quality after 8000 epochs.

default, yet optimal, prioritization of the loss components as specified in Table 4. The incorporation of mirage infusion notably enhances both stability and novelty; however, it also extends the time necessary to reach peak performance.

## C    EXPERIMENTAL DETAILS

**Code**    A zip archive containing the raw code for all the experiments is accessible for download through this link.

**Dataset**    All experiments were conducted using the MP-20 dataset (Jain et al., 2013a), which comprises 45,231 stable inorganic materials selected from Material Projects (Jain et al., 2013a). We employed the same train-validation-test split of 60-20-20 as used in the study by Xie et al. (2022).

**Neural network architecture**    The models utilized the CSPNet neural network architecture proposed by Jiao et al. (2024a), with the following hyperparameters: "hidden_dim" : 512, "num_layers" : 6, "num_freqs" : 128, "latent_dim" : 256, "max_atoms" : 100, "act_fn" : "silu", "dis_emb" : "sin", "edge_style" : "fc", "max_neighbors" : 20, "cutoff" : 7.0, "ln" : True, "ip" : True. The application of mirage infusion, which necessitates the use of an additional atom type 0 in the D3PM diffusion module corresponding to atom types (refer to Sections 2.3.1 and 3), does not increase the number of neural network parameters. This is because the configuration proposed by Jiao et al. (2024a) already includes several free atom types that can be utilized in the proposed model.

**Optimization**    Jiao et al. (2024a) proposed utilizing a batch size of 256 in conjunction with the Adam optimizer (Kingma & Ba, 2017) with an initial learning rate of $10^{-3}$ and a scheduler that reduces the learning rate to $10^{-4}$ when the validation loss ceases to decrease. We use the same batch size, but in contrast, we employed the Adam optimizer solely with a learning rate of $10^{-3}$, omitting any schedulers. Our experiments indicated that the validation loss is not strongly correlated with the quality of the diffusion model once the model performance is sufficiently high. Additional experiments confirmed that this modification in the optimization procedure does not influence the quality of the default DiffCSP. We also found that 1000 epochs were adequate for training the original DiffCSP model, but insufficient for a model after the application of mirage infusion. For the best-proposed version of mirage infusion, maximum performance (as measured by S.U.N. computed via eq-V2) was achieved after 8000 epochs (see Figure 3). Due to this increased requirement for the number of epochs, we removed the scheduler to allow further improvements in the models after the 1000 epochs of training.

**Computational costs**   The proposed mirage infusion technique (see Section 3) substantially increases the average number of atoms with which the neural network interacts during the training and sampling procedures, leading to increased time and memory costs. After applying the mirage infusion technique in its optimal configuration ($N_{\mathrm{m}} = 25$, which corresponds to approximately a $\times 2.7$ increase in the average number of atoms in the crystals), we observe the following approximate increases in execution time: training step $\times 4.3$, sampling step $\times 3.8$; and the following approximate increases in memory consumption: training step $\times 4.4$, sampling step $\times 4.6$. These estimates were obtained on a single GPU NVIDIA Tesla V100 32 GB with 4 CPU cores Intel Xeon Gold 6152 (2.1–3.7 GHz). In our experiments, MiAD ($N_{\mathrm{m}} = 25$) required 4 days for the training procedure (8000 epochs) and 2 hours for the sampling procedure (10 000 crystals) on 2 GPUs NVIDIA Tesla A100 80 GB with 8 CPU cores AMD EPYC 7702 2-3.35 HHz. The training and sampling procedures can also be conducted on a single GPU, though this will incur increased time costs.

## D   DFT COMPUTATION DETAILS

We use DFT settings from Materials Project `https://docs.materialsproject.org/methodology/materials-methodology/calculation-details/gga+u-calculations/parameters-and-convergence` for structure relaxation and energy computation. In particular, we do GGA and GGA+U calculations with `atomate2.vasp.flows.mp.MPGGADoubleRelaxStaticMaker` (Ganose et al., 2025), which in turn relies on `pymatgen.io.vasp.sets.MPRelaxSet` and `pymatgen.io.vasp.sets.MPStaticSet` (Ong et al., 2013). Computations themselves were done with VASP (Kresse & Furthmüller, 1996) version 5.4.4. The raw total energies computed by DFT were corrected with `MaterialsProject2020Compatibility` before putting into the `PhaseDiagram` to obtain the DFT $E^{\mathrm{hull}}$. We used the MP convex hull `2023-02-07-ppd-mp.pkl.gz` distributed by `matbench-discovery` (Riebesell et al., 2023) as the reference hull.

## E   ADDITIONAL METRICS

While S.U.N. remains the principal criterion for evaluating modern generative models in de novo crystal generation Miller et al. (2024); Sriram et al. (2024); Joshi et al. (2025), it is instructive to consider additional metrics that are being used in the literature. These include Structure Validity, Compositional Validity, Coverage (COV-R, COV-P), and distributional distances such as the Wasserstein distance of scalar material properties (e.g., density, number of elements), proposed in Xie et al. (2022). Such metrics provide complementary perspectives on model performance, quantifying local consistency of atomic structures or alignment with empirical property distributions.

However, these measures exhibit important limitations. Particularly, Validity and Coverage are nearly saturated for modern models, as shown in Table 6, with reported differences between approaches often below 1%. More critically, they fail to penalize overfitting. For example, if the MP-20 training set is itself evaluated as a "generative model", it achieves near-optimal values across Validity and Coverage, underscoring that such metrics can be artificially inflated just by replication of training data. Consequently, a model that prioritizes memorization over discovery may appear competitive according to these statistics, despite offering little value for materials discovery.

An illustrative case can be drawn by comparing MiAD with FlowLLM. FlowLLM reports a higher Compositional Validity (89.05% vs. 84.21%), yet its Uniqueness & Novelty (U&N) rate shown in Table 1 is markedly lower (33.8% vs. 65.2%). The discrepancy arises because FlowLLM predominantly reproduces training set compositions, which maximizes apparent validity while suppressing novelty. In contrast, MiAD sacrifices a small degree of compositional accuracy but generates a significantly larger proportion of genuinely new materials, which is the central objective of de novo generation.

To further validate the proposed MiAD approach, we conducted the experiments with small-scale datasets, Perov-5 and Carbon-24. For both of them, we employed the same mirage infusion configuration as for MP-20, specifically $N_{\mathrm{m}} = N + 5$, where $N$ is the maximum number of atoms in crystals in each respective dataset (i.e., $N_{\mathrm{m}} = 10$ for Perov-5 and $N_{\mathrm{m}} = 29$ for Carbon-24). All

Table 6: Evaluation of MiAD on Perov-5, Carbon-24, and MP-20 datasets compared with baseline models. All results follow the same mirage infusion configuration as for MP-20.

| Data | Method | Validity (%)↑ | | Coverage (%)↑ | | Property (%)↓ | |
| | | Struc. | Comp. | COV-R | COV-P | $d_\rho$ | $d_{\text{elem}}$ |
| --- | --- | --- | --- | --- | --- | --- | --- |
| Perov-5 | DiffCSP | 100.00 | 98.85 | 99.74 | 98.27 | 0.111 | 0.013 |
| | CrysBFN | 100.00 | 98.86 | 99.52 | 98.63 | 0.073 | 0.010 |
| | TGDMat | 100.00 | 98.63 | 99.83 | 99.52 | 0.050 | 0.009 |
| | MiAD | 94.82 | 97.91 | 98.07 | 92.82 | 0.089 | 0.075 |
| Carbon-24 | DiffCSP | 100.00 | NA | 99.90 | 97.27 | 0.081 | NA |
| | CrysBFN | 100.00 | NA | 99.90 | 99.12 | 0.061 | NA |
| | TGDMat | 100.00 | NA | 99.99 | 92.43 | 0.043 | NA |
| | Uni-3DAR | 99.99 | NA | 100.00 | 98.16 | 0.066 | NA |
| | MiAD | 99.85 | NA | 99.51 | 99.46 | 0.061 | NA |
| MP-20 | DiffCSP | 100.00 | 83.25 | 99.71 | 99.76 | 0.350 | 0.340 |
| | UniMat | 97.20 | 89.40 | 99.80 | 99.70 | 0.088 | 0.056 |
| | FlowMM | 96.85 | 83.19 | 99.49 | 99.58 | 0.239 | 0.083 |
| | FlowLLM | 99.81 | 89.05 | 99.06 | 99.68 | 0.660 | 0.090 |
| | SymmCD | 90.34 | 85.81 | 99.58 | 97.76 | 0.230 | 0.400 |
| | WyFormer+DiffCSP++ | 99.80 | 81.40 | 99.51 | 95.81 | 0.360 | 0.079 |
| | SymmBFN | 94.27 | 83.93 | 99.73 | 99.00 | 0.083 | 0.095 |
| | CrysBFN | 100.00 | 87.51 | 99.09 | 99.79 | 0.207 | 0.163 |
| | TGDMat | 100.00 | 92.97 | 99.89 | 99.95 | 0.338 | 0.289 |
| | Uni-3DAR | 99.89 | 90.31 | 99.62 | 99.83 | 0.477 | 0.069 |
| | MiAD | 99.25 | 84.21 | 99.35 | 99.80 | 0.233 | 0.027 |
| | MP-20 Train | 100.00 | 90.65 | 99.81 | 99.79 | 0.133 | 0.025 |

other hyperparameters (batch size, number of epochs, and neural network architecture) were adopted directly from DiffCSP.

A comparison of models trained on the Perov-5 and Carbon-24 datasets using the S.U.N. metric is not meaningful because both datasets contain materials that are thermodynamically unstable under standard conditions Xie et al. (2022). Consequently, recent studies Zeni et al. (2024); Miller et al. (2024); Sriram et al. (2024); Joshi et al. (2025); Kazeev et al. (2025) have discontinued the use of these two datasets for benchmarking de novo generation. Currently, there are no well-established methods for fair comparison on these datasets that penalize overfitting. Therefore, Validity, Coverage, and Property could be used only to validate the model's ability to generate coherent structures, while ignoring overfitting. In this light, Table 6 shows that MiAD is competitive with the prior baselines.

This analysis highlights why S.U.N. remains the most informative single metric. Unlike isolated measures, S.U.N. directly evaluates the trade-off between replicating known structures and discovering stable, previously unobserved crystals, thereby capturing the utility of generative models in the materials discovery pipeline. While auxiliary metrics may still be valuable for diagnostic purposes, they should be interpreted with caution, as they may obscure or even contradict the overarching goal of novelty-driven generation.

## F  Distribution of number of atoms

Mirage infusion incorporates the ability to insert and remove atoms during the generation process. However, we must verify that the model actually learns to use this capability. To this end, we report statistics on the number of atoms in S.U.N. crystals generated by MiAD (see Figure 4).

$p(A_{T,i}) = \text{Cat}(A_{T,i} \mid \mathbf{1}/(N_{\text{types}} + 1))$ — atom types (including the mirage type) have a uniform prior (see Section 2.3.1)), where $N_{\text{types}} = 100$ in practice. Thus, if we use mirage infusion with $N_m = 25$, then at the start of generation ($t = T$) all atoms in a crystal are real with probability $\approx 0.77$, exactly

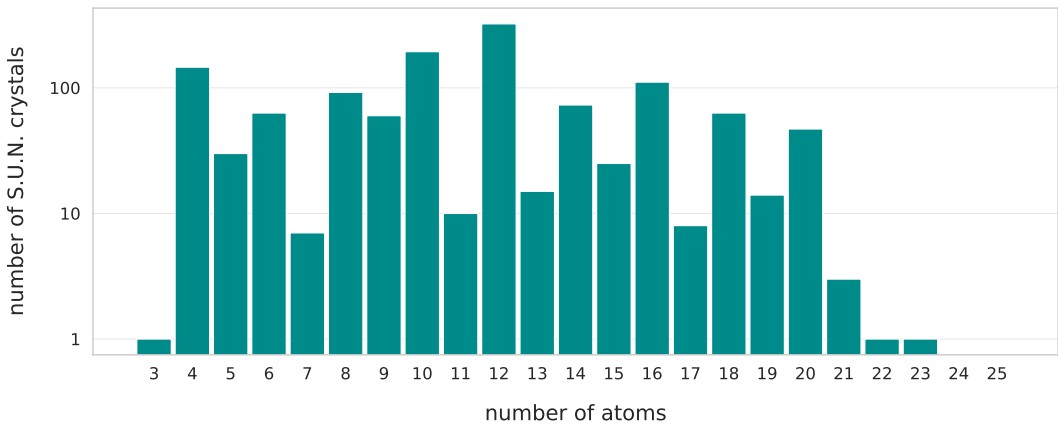

Figure 4: Number of atoms in S.U.N. crystals generated by MiAD. We consider only S.U.N. crystals among the 10 000 crystals generated by MiAD. After generation, crystals are prerelaxed via CHGNet. Stability is also estimated via CHGNet.

one atom is mirage with probability $\approx 0.19$, and so on. This implies that, at the start of generation, almost all crystals contain more than 23 real atoms. This fact, together with the statistics in Figure 4, demonstrates that MiAD

- changes the number of atoms in a crystal during generation;

- generates S.U.N. crystals with different numbers of atoms.

Crystals with 3, 7, 11, 13, 17 atoms appear more rarely than others, because crystals with these atom counts are underrepresented in the training data.

## G    SPACEGROUP DISTRIBUTION

Diversity is a key aspect of the quality of a generative model. S.U.N. addresses this aspect via U – uniqueness. However, this is not the only relevant evaluation, because crystals can differ from one another in various ways. At the same time, many existing diffusion-like models (as well as MiAD) for crystal generation cannot guarantee that the generated crystals will span different spacegroups (particularly the more complex ones, such as cubic or hexagonal). The common assumption is that a diffusion model will learn this from the training data.

To address these concerns, we compare the number of S.U.N. crystals belonging to each spacegroup among 10 000 crystals generated by different models and prerelaxed using CHGNet (see Figure 5). Based on this experiment, we can derive the following claims:

- Existing generative models for crystals do not suffer from severe mode collapse;

- MiAD, on average, generates more S.U.N. crystals than other models without exhibiting mode collapse.

From the application perspective, we argue that the particular shape of the spacegroup distribution is not one of the primary aspects of generative model quality in materials discovery. A spacegroup distribution that is closest to the train or test distribution, or that is the most uniform, does not by itself indicate clear benefits of a particular generative model for the task of inventing novel materials. At the same time, the shape of such distributions can indicate drawbacks if the generative model completely ignores crystals from a particular spacegroup. Our main point is that we should not conflate distribution-based quality criteria with S.U.N., but rather use

- S.U.N. as an averaged quality measure;

- distribution-based metrics as diagnostics to check for the absence of serious diversity issues.

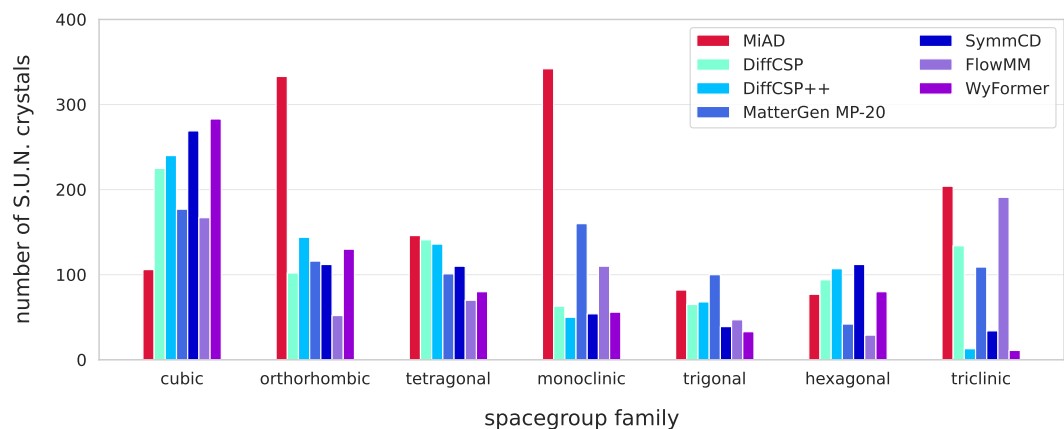

Figure 5: Comparison of models for de novo crystal generation in terms of numbers of S.U.N. crystals belonging to main spacegroup families. We categorize only S.U.N. crystals produced by generative models with a fixed budget of $10\,000$ generated crystals. After generation, crystals are prerelaxed via CHGNet. Stability is also estimated via CHGNet. Spacegroup families are identified via `SpacegroupAnalyzer` from the `pymatgen` package with tolerance: $0.1$.

Table 7: **Comparison of MiAD and MatterGen trained on Alex-MP20** We report stability and S.U.N. estimated eq-V2 for $10\,000$ sampled crystals. For clarity in evaluating the model's quality, we also report the Unique&Novel rate among stable crystals.

| Model | eq-V2 ($E^{\text{hull}} < 0.0$) | | |
| --- | --- | --- | --- |
| | Stable (%) ↑ | Unique&Novel (%) ↑ | S.U.N. (%) ↑ |
| MatterGen | 5.2 | **39.4** | 2.1 |
| MiAD | **7.1** | 35.3 | **2.5** |

## H  COMPARISON ON ALEX-MP20

To demonstrate the scalability of MiAD, we conduct experiments on the large-scale Alex-MP20 dataset (Zeni et al., 2024) and compare its performance with that of MatterGen. We employ S.U.N., estimate stability using eq-V2, and use the same energy hull as in the previous experiments. The only difference in the evaluation protocol is that novelty is measured relative to the Alex-MP20 training set. Samples from both models are pre-relaxed for 100 steps using eq-V2. MiAD is trained for 1200 epochs with the same hyperparameters as in the MP-20 experiments. The results in Table 7 demonstrate that MiAD, without additional hyperparameter tuning, achieves state-of-the-art performance on one of the largest datasets for de novo crystal generation.

## I  LLM USAGE

The text of the paper was polished for grammar and style using LLMs.

## J  QUALITATIVE ANALYSIS OF MIRAGE INFUSION DYNAMICS

The proposed MiAD framework generalizes standard crystal diffusion models by introducing dynamic atom counts. This approach differs from fixed-size baselines along three primary axes: (1) the formulation of the generative task, (2) the robustness of generation trajectories, and (3) the expanded action space available during diffusion. While the first two axes are implicit, the third, the ability to dynamically add or remove atoms, enables a direct analysis of how the model manages structural evolution. In this section, we investigate the hypothesis that mirage atoms provide an "error correction" mechanism, allowing the model to recover from unstable intermediate states.

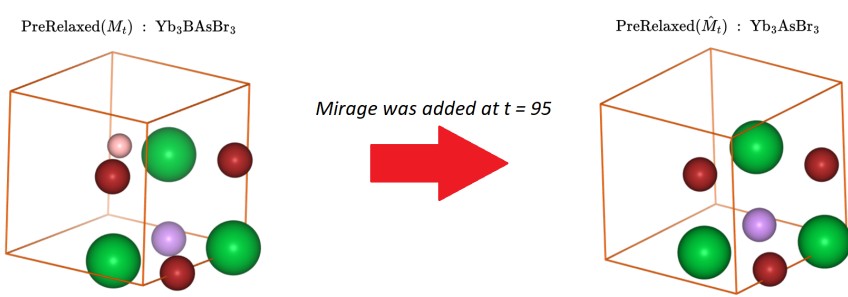

PreRelaxed($M_t$) : $Yb_3BAsBr_3$

*Mirage was added at t = 95*

PreRelaxed($\hat{M}_t$) : $Yb_3AsBr_3$

Figure 6: Comparison between the counterfactual structure $M_t$ (left), and the corrected structure $\hat{M}_t$ (right). The corrected structure is thermodynamically stable with a higher symmetry group (monoclinic) compared to the unstable, triclinic counterfactual.

## J.1 METHODOLOGY

To isolate the impact of the mirage mechanism, we focused on "boundary" cases in the generation trajectories where the model converts a real atom into a mirage atom (effectively removing it) during the final stages of the reverse diffusion process. We performed a comparative analysis on these samples, defined as follows:

**The Counterfactual** ($M_t$): The structure at the moment right before the last change in the denoising trajectory from a real atom to a mirage atom. In this state, the atom remains a real element, simulating a fixed-size model that cannot delete atoms, even when they are structurally disadvantageous.

**The Corrected Structure** ($\hat{M}_t$): The crystal structure with the same set of atoms as right after the model successfully changed the real atom to the mirage atom. To ensure that differences arise solely from the composition change (and not from stochastic noise or drift), we preserve the exact lattice parameters and fractional coordinates of all remaining atoms from $M_t$.

Both $M_t$ and $\hat{M}_t$ were subsequently pre-relaxed using CHGNet to evaluate their stability and symmetry properties.

## J.2 EVIDENCE OF ERROR CORRECTION

Our observations reveal that the mirage mechanism functions as a dynamic pruning tool. Out of 1000 generated crystals, we analyzed a subset of 57 that were identified as Stable, Unique, and Novel, where the transition from a real atom to a mirage occurred after 150 steps of the denoising process (out of 1000).

The removal of the superfluous atoms in these trajectories resulted in drastic improvements in thermodynamic stability:

1. The median energy above hull ($E^{\text{hull}}$) dropped from $0.29$ eV/atom for the counterfactuals ($M_t$) to $0.033$ eV/atom for the corrected structures ($\hat{M}_t$).

2. The median absolute deviation of the energy narrowed significantly from $0.27$ to $0.12$, indicating a more consistent convergence toward stable minima.

Furthermore, the structural symmetry improved substantially. Among the 57 crystals, only 14 exhibited non-trivial symmetry groups (cubic, monoclinic, orthorhombic, triclinic, or trigonal) in the uncorrected state. After the mirage infusion, this number nearly doubled to 27 crystals. This suggests that the model effectively removes atoms that break symmetry or disrupt the lattice, a capability structurally impossible for fixed-size diffusion baselines.

Table 8: **Scaling mirage infusion to MPTS-52** MiAD is trained with $N_m = 57$. MiAD-WRNA is trained with $N_m = N_{atoms} + k$, where $k \sim \mathcal{U}[0, 10]$ (+5 mirage atoms in average). The models are compared using S.U.N. for 10 000 sampled crystals, where stability is estimated via (left) eq-V2 and (right) CHGNet.

| Model | eq-V2 ($E^{hull} < 0.0$) | | | CHGNet ($E^{hull} < 0.0$) | | |
|---|---|---|---|---|---|---|
| | Stable (%) ↑ | Unique&Novel (%) ↑ | S.U.N. (%) ↑ | Stable (%) ↑ | Unique&Novel (%) ↑ | S.U.N. (%) ↑ |
| DiffCSP | 4.3 | 37.2 | 1.6 | 11.8 | 69.9 | 8.2 |
| MiAD | 5.7 | 55.6 | 3.2 | 13.0 | 76.2 | 9.9 |
| MiAD-WRNA | 5.6 | 44.1 | 2.5 | 13.0 | 70.9 | 9.2 |

### J.3 CASE STUDY

A concrete example illustrates this behavior: in one trajectory(see Figure 6), the model removed a Boron atom at step 95 (of 1000), shifting the composition from $Yb_3BAsBr_3$ ($M_t$) to $Yb_3AsBr_3$ ($\hat{M}_t$). While both structures initially possessed triclinic symmetry, the pre-relaxed crystal $\hat{M}_t$ converged to a higher-symmetry monoclinic group and achieved thermodynamic stability (negative energy above hull, $-0.083$ eV/atom). Conversely, the counterfactual $M_t$ failed to find higher symmetry, remaining triclinic, and was found to be heavily unstable (0.335 eV/atom).

This confirms that the mirage atoms allow the model to alleviate mistakes made during earlier stages of generation, dynamically refining the composition to ensure realizability.

## K SCALABILITY ON CRYSTALS WITH LARGER NUMBER OF ATOMS

Mirage infusion enables a joint diffusion model operating on the $(L, F, A)$ crystal representation to change the number of atoms in a crystal during generation. To assess how this procedure scales when the admissible range of atom counts is wider, we conduct experiments on MPTS-52 – a more challenging extension of MP-20 (Jain et al., 2013a), comprising 40,476 structures with up to 52 atoms per cell, ordered by earliest publication year in the literature.

There are several challenges in conducting comparisons on MPTS-52. To our knowledge, prior works have not reported wide comparisons of S.U.N. metrics for different generative models on MPTS-52. Metrics such as Structure Validity, Compositional Validity, Coverage (COV-R, COV-P), and distributional distances (e.g., the Wasserstein distance of scalar material properties such as density or number of elements) proposed in (Xie et al., 2022) have significant limitations, which we discuss in Appendix E.

It is important to consider the DiffCSP backbone architecture, which operates most effectively with crystals containing fewer than 20 atoms. Applying DiffCSP to MPTS-52 places the model in a regime where performance degrades due to the larger number of atoms. Applying mirage infusion further increases the number of atoms; thus, when we use the same network architecture, that challenge is amplified.

We trained the original DiffCSP and MiAD on MPTS-52 for 6k epochs (six times the number of epochs used in the original DiffCSP for MPTS-52) and selected the best checkpoints by S.U.N. We use the same neural network as in experiments on MP-20. We did not tune MiAD for MPTS-52, but instead used the configuration identified as optimal on MP-20:

**MP-20:** $N_m = \text{maximum\_number\_of\_atoms\_in\_mp20} + 5 = 20 + 5 = 25$

**MPTS-52:** $N_m = \text{maximum\_number\_of\_atoms\_in\_mpts52} + 5 = 52 + 5 = 57$

The results are presented in Table 8. Here, we observe consistent improvements with mirage infusion using the same configuration (without finding optimal hyperparameters for dataset). Computational overhead: 1 epoch of DiffCSP training on MPTS-52: 35 sec, while 1 epoch of MiAD ($N_m = 57$) training on MPTS-52: 79 sec ($\approx 2.26$x compared to DiffCSP). The overhead is smaller than in MP-20 (see Appendix C) because the network is already near its upper limit of effective atomic interactions (20 atoms); further increases in atom count then lead to approximately linear, not quadratic, scaling.

Applying mirage infusion to datasets with a broad range of atom counts is a plausible use case in practical applications. For such datasets, we can vary the number of mirage atoms added per structure while keeping the rest of the procedure unchanged. We demonstrate this variant on MPTS-52. Specifically, we set $N_{\mathrm{m}}$ per crystal to $N_{\mathrm{atoms}} + \mathrm{Uniform}[0, 10]$, i.e., each crystal receives 0-10 mirage atoms (+5 on average). At generation start, we sample the number of atoms from the training-set distribution (as in the original DiffCSP), and add a sample from $\mathrm{Uniform}[0, 10]$. The results for this variant (MiAD-WRNA: MiAD for Wide Ranges of Number of Atoms) presented in Table 8. MiAD-WRNA still provides substantial gains over DiffCSP, although it performs slightly worse than the original MiAD with fixed $N_{\mathrm{m}}$. Its main advantage lies in computational cost: 1 epoch of DiffCSP training on MPTS-52: 35 sec; 1 epoch of MiAD ($N_{\mathrm{m}} = 57$) training on MPTS-52: 79 sec ($\approx 2.26$x vs DiffCSP); 1 epoch of MiAD-WRNA (+5 avg) training on MPTS-52: 40 sec ($\approx 1.14$x vs DiffCSP). Thus, MiAD-WRNA offers a practical trade-off between computational cost and quality.

These MPTS-52 experiments, together with Table 3, support the effectiveness of mirage atoms across different strategies for incorporating them into crystals.

