# OpenReview forum: "MiAD: Mirage Atom Diffusion for De Novo Crystal Generation"
_ICLR.cc/2026/Conference — Submitted to ICLR 2026_

### Official Review · Reviewer_ixHC · 2025-10-27

**Soundness:** 4
**Presentation:** 3
**Contribution:** 1
**Rating:** 4
**Confidence:** 5

**Summary:**

The paper proposes adding mirage (non-existent) atoms to crystal systems to enable diffusion models to modify the number of atoms during inference, thus removing the restriction to pre-specify the number of atoms for crystal generation. The method, MiAD, generates significantly more stable, unique, and novel (S.U.N.) crystals than the baselines.

**Strengths:**

- The paper is well-structured and written.
- The proposed idea is simple with minor modifications to existing diffusion model training pipelines.
- The method notably achieves higher S.U.N. metrics for crystal generation than the compared baselines.

**Weaknesses:**

I primarily have concerns about:
- lack of some appropriate experiments or discussions about design choices
- lack of conceptual discussions about the MiAD method

These are supported by the questions below. I am willing to improve the score if these are adequately addressed during the discussion period.

**Questions:**

- What is the appropriate method to select $N_m$? For MP-20, S.U.N. performance degrades (and then somewhat increases?) when $m$ exceeds 25 (if finer choices are used, performance could be better at other values as well, and since the method is quite sensitive to this hyperparameter, this is a concern). Is there any explanation for this? Is there a general rule of thumb about how much $m$ should be, if the maximum number of atoms in the training dataset is $N_d$?
- Can you provide the results of MiAD on datasets with a larger number of atoms, such as MPTS-52? This would show if mirage infusion and reduction also improve performance in such crystals.
- Conceptually, with the addition of mirage atoms, the distribution shifts from the original training distribution (and potentially changes the symmetry within the crystal). Does the model have to learn two symmetries simultaneously? Is there a study on the distribution of space groups before and after mirage infusion, along with the generated crystals before and after reduction?
- It would be great to add more discussion on the benefits of your method compared to the first subgroup of methods (under models that change the number of atoms) to highlight the importance of the proposed method in developing models that can change the number of atoms during inference. Furthermore, the result tables lack these baselines, which are more relevant for comparison with MiAD.
- The mirage infusion and reduction are implemented only with DiffCSP. Results with additional models that require no special modification (e.g., as mentioned, FlowMM, CrystalFlow, and MatterGen) will demonstrate the method's adaptability and whether performance improvements are agnostic to the underlying model.
- What is the quantitative estimate of the increase in training and inference budget (time, compute, memory) with the mirage atoms (since it is mentioned that the model incurs higher computational costs)?

---

> ### Author Response · Authors · 2025-11-25
>
> Below we provide a point-by-point response to the reviewer’s questions and concerns. We appreciate the thoughtful feedback and the opportunity to clarify our design choices, empirical findings, and conceptual framing. Wherever possible, we reference tables and figures in the submission and provide concrete quantitative details. Our overarching goal is to make clear how MiAD behaves across settings, why certain hyperparameters matter, and how our method relates to prior approaches that support variable atom counts. We will incorporate the clarifications and improvements outlined here in a revised version.
>
> - “What is the appropriate method to select $N_{m}$? For MP-20, S.U.N. performance degrades (and then somewhat increases?) when $m$ exceeds 25 (if finer choices are used, performance could be better at other values as well, and since the method is quite sensitive to this hyperparameter, this is a concern). Is there any explanation for this? Is there a general rule of thumb about how much $m$ should be, if the maximum number of atoms in the training dataset is $N_{d}$?”
>
> We begin by noting the observation from Table 3 that using $N_{\text{m}}$ = 20 or 25 both provides a substantial boost relative to the original DiffCSP:
>
> -> DiffCSP: SUN (eq-V2) = 2.5\%, SUN (CHGNet) = 8.9\%
>
> -> MiAD (20): SUN (eq-V2) = 5.3\%, SUN (CHGNet) = 12.0\%
>
> -> MiAD (25): SUN (eq-V2) = 5.5\%, SUN (CHGNet) = 12.9\%
>
> Furthermore, considering the variability in S.U.N. values across “neighboring” checkpoints (Figure 2, page 7) -- for example, between 6k and 8k epochs or between 8k and 10k epochs -- the difference between MiAD (20) and MiAD (25) is sufficiently small. MiAD with $N_{\text{m}}$ = 20 essentially enables changing the number of atoms. Choosing larger $N_{\text{m}}$ values (e.g., 25, 30, 35) increases the set of atom variants among which the model can choose, which can have a positive effect. However, the neural network inside DiffCSP can operate at peak efficacy primarily on crystals with no more than 20 atoms. This constraint may limit further gains.
>
> Our current conclusions from the MP-20 experiments are as follows:
>
> RULE: Choosing $N_{\text{m}}$ equal to the maximum number of atoms $N$ in the crystals, or slightly greater, works well. Further increasing $N_{\text{m}} > N$, once it exceeds the atom count with which the neural network can effectively operate, may lead to a degradation in quality.
>
> Other rules may emerge depending on the distribution of atom counts in the dataset and on the neural architecture. For instance, consider a synthetic dataset where all crystals have fewer than 20 atoms except for a single crystal with 100 atoms. The model’s behavior on such a dataset would be unpredictable because there are no intermediate examples between 20 and 100 atoms. However, if the distribution of atom counts does not have large regions of zero probability (as in the 20-100 gap in the synthetic example), we expect the above RULE to hold.

---

> > ### Author Response · Authors · 2025-11-25
> >
> > - “Can you provide the results of MiAD on datasets with a larger number of atoms, such as MPTS-52? This would show if mirage infusion and reduction also improve performance in such crystals.”
> >
> > There are several challenges in conducting comparisons on MPTS-52. To our knowledge, prior works have not reported S.U.N. metrics for generative models on MPTS-52, and therefore we did not include such experiments in the initial version of the paper. Metrics such as Structure Validity, Compositional Validity, Coverage (COV-R, COV-P), and distributional distances (e.g., the Wasserstein distance of scalar material properties such as density or number of elements) proposed in Xie et al. (2022) have significant limitations, which we discuss in the appendix Additional metrics, page 19. Nevertheless, we are able to compare the original DiffCSP with MiAD to provide additional evidence of the effectiveness of the proposed methodology.
> >
> > It is important to consider the DiffCSP backbone architecture, which operates most effectively with crystals containing fewer than 20 atoms. Applying DiffCSP to MPTS-52 places the model in a regime where performance degrades due to the larger number of atoms. Applying mirage infusion further increases the number of atoms; thus, when we use the same network architecture, that challenge is amplified.
> >
> > We trained the original DiffCSP and MiAD on MPTS-52 for 6k epochs (six times the number of epochs used in the original DiffCSP for MPTS-52) and selected the best checkpoints by S.U.N. We use the same neural network as in experiments on MP-20. We did not tune MiAD for MPTS-52, but instead used the configuration identified as optimal on MP-20:
> >
> > -> For MP-20: $N_{\text{m}}$ = maximum\_number\_of\_atoms\_in\_mp20 + 5 = 20 + 5 = 25
> >
> > -> For MPTS-52: $N_{\text{m}}$ = maximum\_number\_of\_atoms\_in\_mpts52 + 5 = 52 + 5 = 57
> >
> > The results are as follows (S - Stability rate $E^{hull} < 0$, UN - Unique and Novel among Stable rate):
> >
> > | Name | S (eq-V2) | UN | S.U.N. (eq-V2) | S (CHGNet) | UN | S.U.N. (CHGNet)|
> > |---|---|---|---|---|---|---|
> > | DiffCSP | 4.27 | 37.24 | 1.59 | 11.76 | 69.90 | 8.22 |
> > | MiAD | 5.72 | 55.59 | 3.18 | 13.04 | 76.15 | 9.93 |
> >
> > Here, we observe consistent improvements with mirage infusion using the same configuration (without finding optimal hyperparameters for dataset).
> >
> > Computational overhead:
> >
> > -> 1 epoch of DiffCSP training on MPTS-52: 35 sec
> >
> > -> 1 epoch of MiAD ($N_{\text{m}} = 57$) training on MPTS-52: 79 sec ($\approx 2.26$x compared to DiffCSP)
> >
> > The overhead is smaller than in MP-20 (appendix Experimental details, paragraph Computational costs, page 19) because the network is already near its upper limit of effective atomic interactions (20 atoms); further increases in atom count then lead to approximately linear, not quadratic, scaling.
> >
> > Applying mirage infusion to datasets with a broad range of atom counts is a plausible use case in practical applications. For such datasets, we can vary the number of mirage atoms added per structure while keeping the rest of the procedure unchanged. We demonstrate this variant on MPTS-52.
> >
> > Specifically, we set $N_{\text{m}}$ per crystal to $N_{\text{atoms}} + \text{Uniform}[0, 10]$, i.e., each crystal receives 0–10 mirage atoms (+5 on average). At generation start, we sample the number of atoms from the training-set distribution (as in the original DiffCSP), and add a sample from Uniform$[0, 10]$. The results for this variant (MiAD-WRNA: MiAD for Wide Ranges of Number of Atoms) are (S - Stability rate $E^{hull} < 0$, UN - Unique and Novel among Stable rate):
> >
> > | Name | S (eq-V2) | UN | S.U.N. (eq-V2) | S (CHGNet) | UN | S.U.N. (CHGNet)|
> > |---|---|---|---|---|---|---|
> > | DiffCSP | 4.27 | 37.24 | 1.59 | 11.76 | 69.90 | 8.22 |
> > | MiAD | 5.72 | 55.59 | 3.18 | 13.04 | 76.15 | 9.93 |
> > | MiAD-WRNA | 5.55 | 44.14 | 2.45 | 13.03 | 70.91 | 9.24 |
> >
> > MiAD-WRNA still provides substantial gains over DiffCSP, although it performs slightly worse than the original MiAD with fixed $N_{\text{m}}$. Its main advantage lies in computational cost:
> >
> > -> 1 epoch of DiffCSP training on MPTS-52: 35 sec
> >
> > -> 1 epoch of MiAD ($N_{\text{m}} = 57$) training on MPTS-52: 79 sec ($\approx 2.26$x vs DiffCSP)
> >
> > -> 1 epoch of MiAD-WRNA (+5 avg) training on MPTS-52: 40 s ($\approx 1.14$x vs DiffCSP)
> >
> > Thus, MiAD-WRNA offers a practical trade-off between computational cost and quality. These MPTS-52 experiments, together with Table 3, support the effectiveness of mirage atoms across different strategies for incorporating them into crystals. We will include these results in future revisions of the paper.

---

> > > ### Author Response · Authors · 2025-11-25
> > >
> > > - “Conceptually, with the addition of mirage atoms, the distribution shifts from the original training distribution (and potentially changes the symmetry within the crystal). Does the model have to learn two symmetries simultaneously? Is there a study on the distribution of space groups before and after mirage infusion, along with the generated crystals before and after reduction?”
> > >
> > > Yes. Mirage infusion changes the training data distribution relative to the original distribution. In particular, mirage atoms are inserted at random positions, so the resulting partially randomized crystal structure has approximately zero probability of being symmetric. Consequently, the symmetries pertain only to atoms of a given type, and the model learns this structure successfully. In appendix Spacegroup distribution, Figure 5, page 22, we present the distribution of space groups among S.U.N. crystals and observe that crystals generated by MiAD exhibit symmetries after the mirage atoms are removed. If we retain the mirage atoms and plot the same histogram, all crystals lack symmetry and fall into the triclinic space group (we confirmed this experimentally, although these results are not included in the paper).
> > >
> > > - “It would be great to add more discussion on the benefits of your method compared to the first subgroup of methods (under models that change the number of atoms) to highlight the importance of the proposed method in developing models that can change the number of atoms during inference. Furthermore, the result tables lack these baselines, which are more relevant for comparison with MiAD.”
> > >
> > > We are happy to provide this analysis here. We had a version in the main text but removed it due to space limitations.
> > >
> > > These approaches modify the representation of 3D crystal structures to allow changing the number of atoms during generation:
> > >
> > > -> Uni-3DAR proposes an autoregressive model over a hierarchical tokenized representation that compresses 3D space using an octree. The model progressively constrains the unit-cell space via a sequence of levels (sets of cubes), where each level refines the previous one. Autoregression enables the model to generate a variable number of cubes at each level, thereby choosing the number of atoms in the unit cell; the hierarchical representation decomposes the generation into multiple steps. Uni-3DAR differs substantially from MiAD because the autoregressive paradigm is inherently suited to variable-sized objects.
> > >
> > > -> WyckoffDiff constructs a diffusion model over discrete symmetry-based crystal descriptors. An analogue of mirage atoms appears as atoms in unconstrained Wyckoff positions, which can, in principle, be occupied by any number of atoms -- although in practice there is a maximum. In WyckoffDiff, these “virtual atoms” arise from the choice of representation, not explicitly to enable variable atom counts. A key difference from MiAD is that, in WyckoffDiff, the positions where such atoms can appear during generation are highly constrained by the representation.
> > >
> > > -> UniMat introduces a representation combining periodic-table positioning of atom types with geometric crystal information and applies a standard DDPM in this space. Atoms are added to cells of the periodic table, with empty cells encoded via special coordinates. These empty cells permit the model to create new atoms during generation. Key differences from MiAD include:
> > >
> > > 1) MiAD enforces rotation, translation, periodicity, and permutation invariances by design, whereas UniMat learns them via augmentation.
> > >
> > > 2) MiAD distinguishes real versus mirage atoms by type, whereas UniMat uses coordinate encodings for (non-)occupancy.
> > >
> > > 3) The number of mirage atoms in MiAD is a freely chosen hyperparameter. In contrast, UniMat, when applied to MP-20, uses at least 900+ “empty” positions and cannot be readily configured with fewer.
> > >
> > > Including these models in our quantitative comparison is currently difficult due to the lack of reported S.U.N. metrics (in any version). At present, reproducing WyckoffDiff and UniMat for comparison is not feasible (UniMat lacks an open-source implementation; WyckoffDiff’s public code omits necessary postprocessing). We have not yet attempted to reproduce Uni-3DAR but plan to do so in future work.
> > >
> > > - “The mirage infusion and reduction are implemented only with DiffCSP. Results with additional models that require no special modification (e.g., as mentioned, FlowMM, CrystalFlow, and MatterGen) will demonstrate the method’s adaptability and whether performance improvements are agnostic to the underlying model.”
> > >
> > > We will add these experiments in future revisions and will aim to provide preliminary results during the discussion period if they become available.

---

> > > > ### Author Response · Authors · 2025-11-25
> > > >
> > > > - “What is the quantitative estimate of the increase in training and inference budget (time, compute, memory) with the mirage atoms (since it is mentioned that the model incurs higher computational costs)?”
> > > >
> > > > At the end of the paragraph where computational overhead is mentioned, we refer to appendix Experimental details. There, in paragraph Computational costs, page 19, we provide detailed information:
> > > >
> > > > “The proposed mirage infusion technique (see Section 3) substantially increases the average number of atoms with which the neural network interacts during the training and sampling procedures, leading to increased time and memory costs. After applying the mirage infusion technique in its optimal configuration (Nm = 25, which corresponds to approximately a ×2.7 increase in the average number of atoms in the crystals), we observe the following approximate increases in execution time: training step ×4.3, sampling step ×3.8; and the following approximate increases in memory consumption: training step ×4.4, sampling step ×4.6. These estimates were obtained on a single GPU NVIDIA Tesla V100 32 GB with 4 CPU cores Intel Xeon Gold 6152 (2.1–3.7 GHz). In our experiments, MiAD (Nm = 25) required 4 days for the training procedure (8000 epochs) and 2 hours for the sampling procedure (10 000 crystals) on 2 GPUs NVIDIA Tesla A100 80 GB with 8 CPU cores AMD EPYC 7702 2–3.35 GHz. The training and sampling procedures can also be conducted on a single GPU, though this will incur increased time costs.”
> > > >
> > > > We recognize that the reference from the main text to this appendix content is not sufficiently prominent; we will make this pointer clearer in a future revision.

---

### Official Review · Reviewer_enPF · 2025-10-31

**Soundness:** 2
**Presentation:** 3
**Contribution:** 2
**Rating:** 4
**Confidence:** 5

**Summary:**

In this paper, the authors address a key limitation in crystal material generation, where most existing diffusion-based models are constrained to a fixed number of atoms during the generation process. To overcome this, they propose MiAD (Mirage Atom Diffusion), a novel generative framework that extends DiffCSP by allowing the number of atoms to vary dynamically as the crystal structure evolves. The central idea, called “mirage infusion”, introduces a special placeholder atom type (type 0), referred to as a mirage atom, which can either appear or disappear during generation. This mechanism effectively enables the model to add or remove atoms adaptively, thereby expanding the diversity and flexibility of generated structures. Experimental results on the MP-20 dataset show that MiAD achieves a S.U.N. rate of 8.2%, outperforming leading baselines such as ADiT, WyFormer, and MatterGen, and demonstrating substantial improvements in both generative quality and material discovery potential.

**Strengths:**

- The paper is very well written. The limitation of fixed atom counts in current crystal diffusion models is well identified and practically significant for de novo materials discovery.
- The idea of mirage infusion technique is conceptually simple yet effective, implemented by augmenting the atom-type diffusion process with an additional “mirage” type and masking loss terms appropriately. It Gives the flexibility to the models to vary the number of atoms in a crystal during the generation process.
- Results are compared across several leading baselines (DiffCSP, FlowMM, ADiT, WyFormer, etc.) with DFT-based and MLIP-based stability evaluations. Also, the computational and structural validity results are produced in the appendix.

**Weaknesses:**

- The paper lacks methodological novelty. MiAD’s architecture remains largely identical to DiffCSP, with the only change being the addition of mirage atoms. While this tweak is clever, it is incremental rather than fundamentally new.
- The paper does not convincingly justify why varying atom numbers is scientifically important beyond improving diversity. For instance, how does this help in discovering more stable or experimentally realizable materials?
- The paper could benefit from qualitative visualizations showing how mirage atoms evolve during diffusion, and examples of successful vs failed generations.
- The authors acknowledge higher computational overhead due to an increased average atom count, but there is no quantitative analysis or efficiency comparison.
- Scalability on larger datasets (e.g., MPTS-52) remains unclear. It is not evident how computationally efficient the proposed approach is when incorporating these additional “mirage” atoms at larger scales. A detailed analysis of the computational overhead and efficiency on such datasets would strengthen the work.
- For the S.U.N. comparisons, several important baseline models—such as UniMat[1], TGDMat[2], CrysBFN[3], and Crystal-Text-LLM[4]—are not included. Additionally, key recent baselines like DiffCSP++ and SymmCD are missing from Table 1, while FlowLLM and WyFormer are absent from Table 2. The omission of these models leads to incomplete and inconsistent comparisons, making it difficult to accurately assess the relative performance and claimed improvements of the proposed approach.

[1] Yang, Sherry, et al. "Scalable diffusion for materials generation." arXiv preprint arXiv:2311.09235 (2023).

[2] Das, Kishalay, et al. "Periodic materials generation using text-guided joint diffusion model." ICLR 2025.

[3] Wu, Hanlin, et al. "A periodic Bayesian flow for material generation." ICLR 2025.

[4] Gruver, Nate, et al. "Fine-tuned language models generate stable inorganic materials as text." 2024.

**Questions:**

- Since MiAD’s architecture closely follows DiffCSP, apart from diffusion models have you tested it on other frameworks like LLMs or Flow-based models? Do that show similar performance improvement?
- The paper mentions that varying the number of atoms enhances generative diversity, but could the authors elaborate on its scientific relevance? Specifically, how does this ability contribute to discovering more stable or experimentally realizable materials?
- Can the authors provide qualitative visualizations or case studies showing how mirage atoms evolve during diffusion? For instance, examples of successful and failed generations would help illustrate the behavior and impact of the proposed mechanism.
- Could they provide quantitative comparisons (e.g., training time, memory usage) to clarify the extent of this overhead relative to DiffCSP?
- How scalable is MiAD to larger and more complex datasets such as MPTS-52?
- When incorporating mirage atoms at larger scales, how computationally efficient is the model, and are any optimization strategies employed to maintain tractable training and inference?

---

> ### Author Response · Authors · 2025-11-25
>
> Thank you for the thoughtful and detailed review. We appreciate the careful assessment of our work’s strengths and the constructive feedback on its limitations, presentation, and empirical comparisons. Below, we address each point raised by the reviewer, maintaining the original ideas while clarifying our intentions, improving the academic tone, and correcting stylistic issues. Where appropriate, we also indicate places in the manuscript that will be revised for clarity in future versions. Our goal is to make the contribution, scope, and empirical grounding of the paper as transparent as possible.
>
>
> #### Scientific relevance of varying atom numbers; link to stability and realizability
>
> - Weakness: “The paper does not convincingly justify why varying atom numbers is scientifically important beyond improving diversity. For instance, how does this help in discovering more stable or experimentally realizable materials?”
> - Question: “The paper mentions that varying the number of atoms enhances generative diversity, but could the authors elaborate on its scientific relevance? Specifically, how does this ability contribute to discovering more stable or experimentally realizable materials?”
>
> As noted in the summary, the proposed model is, in theory, a generalization of DiffCSP. However, this does not imply that it will perform better in practice. As shown in appendix Ablation, Table 5, page 17, specific design choices within this generalization can either improve or degrade performance. Therefore, the mere fact that the model is a generalization cannot serve as a justification for improved quality. The question is consequently reduced to whether the particular design choices we adopt lead to performance gains. Non-formal motivation for why we expect such gains is provided in section Method, paragraph Discussion, page 5. A formal proof is not possible in our setting, given that we work with non-ideal neural networks that lack complete knowledge of the space of stable crystals (due to limited data, finite model capacity, etc.). Indeed, if DiffCSP were ideal, our generalization would not yield a quality boost because the baseline would already be optimal. We therefore offer several additional non-formal arguments for why our final approach outperforms DiffCSP.
>
> In broad terms, the proposed diffusion model differs from the prior approach along three axes: (1) the task posed to the neural network (potentially more convenient), (2) the generation trajectories (potentially more robust or simpler to follow), and (3) the set of actions available during diffusion (potentially enabling the model to handle situations that were previously unsolvable during generation). While points (1) and (2) we cannot demonstrate directly and only implicitly support by experimental results, point (3) can be partially observed; we refer to the following question for an indication of this behavior.

---

> ### Author Response · Authors · 2025-11-25
>
> #### Qualitative visualizations of mirage atoms; successful vs. failed generations
>
> - Weakness: “The paper could benefit from qualitative visualizations showing how mirage atoms evolve during diffusion, and examples of successful vs failed generations.”
> - Question: “Can the authors provide qualitative visualizations or case studies showing how mirage atoms evolve during diffusion? For instance, examples of successful and failed generations would help illustrate the behavior and impact of the proposed mechanism.”
>
> To address the request for qualitative insight and case studies, we analyzed specific generation trajectories to understand how the model utilizes mirage atoms to improve structural quality. We focused on "boundary" cases where the model decides to convert an atom into a mirage (effectively removing it) during the final stages of the reverse diffusion process.
>
> We performed a comparative analysis on these samples:
>
> *   **The Counterfactual ($M_{t}$):** The structure at the moment right before the last change in the denoising trajectory from a real atom to a mirage atom. In this state, the atom remains as a real element, simulating a fixed-size model unable to delete atoms.
> *   **The Crystal with final set of atoms ($\hat{M}_{t}$):** The crystal structure with the same set of atoms as right after the model successfully changed the real atom to the mirage atom. We preserve the same lattice and fractional coordinates for all atoms as in $M_{t}$ to ensure the same noise level in $M_{t}$ and $\hat{M}_{t}$.
>
> Both $M_{t}$ and $\hat{M}_{t}$ were pre-relaxed using default settings with CHGNet.
>
> Our observations reveal a distinct "error correction" mechanism supported by quantitative improvements in stability and symmetry. We examined the subset of 57 S.U.N. crystals where the transition from a real atom to a mirage occurred after 150 steps (out of 1000). The removal of the superfluous atom resulted in a drastic improvement in thermodynamic stability: the median energy above hull dropped from 0.29 eV/atom for the counterfactuals ($M_t$) to 0.033 eV/atom for the corrected structures ($\hat{M}_t$), with the median absolute deviation narrowing from 0.27 to 0.12. Furthermore, the number of crystals exhibiting non-trivial symmetry groups (e.g., cubic, monoclinic, orthorhombic, trigonal) nearly doubled—rising from 14 in the uncorrected state to 27 after mirage addition. This indicates that the mirage mechanism allows the model to dynamically "prune" mistakes made earlier in the generation process, removing atoms that disrupt the lattice to recover stability and symmetry—a capability structurally impossible for fixed-size diffusion baselines.
>
> A concrete example illustrates this behavior: In one trajectory, the model removed a Boron atom at step 95 (of 1000), shifting the composition from $Yb_{3} B As Br_{3}$ ($M_t$) to $Yb_{3} As Br_{3}$ ($\hat{M_{t}}$). While both structures initially possessed triclinic symmetry, the pre-relaxed crystal $\hat{M}_{t}$ converged to a higher-symmetry **monoclinic** group and achieved thermodynamic stability (negative energy above hull, -0.083 eV/atom). Conversely, the counterfactual $M_t$ failed to find higher symmetry, remaining triclinic, and was found to be heavily unstable (0.335 eV/atom). We will include these visualizations (https://www.dropbox.com/scl/fi/8om343u6pm5n007r00mf4/MirageRemovalExample.png?rlkey=itajl25ctduhjlzf7279vwl7g&st=rc23itcj&dl=0) in the revised appendix.

---

> > ### Author Response · Authors · 2025-11-25
> >
> > #### Quantitative overhead and efficiency
> >
> > - Weakness: “The authors acknowledge higher computational overhead due to an increased average atom count, but there is no quantitative analysis or efficiency comparison.”
> > - Question: “Could they provide quantitative comparisons (e.g., training time, memory usage) to clarify the extent of this overhead relative to DiffCSP?”
> >
> > At the end of the paragraph where computational overhead is mentioned, we refer to appendix Experimental details. There, in paragraph Computational costs, page 19, we provide detailed information:
> >
> > “The proposed mirage infusion technique (see Section 3) substantially increases the average number of atoms with which the neural network interacts during the training and sampling procedures, leading to increased time and memory costs. After applying the mirage infusion technique in its optimal configuration (Nm = 25, which corresponds to approximately a ×2.7 increase in the average number of atoms in the crystals), we observe the following approximate increases in execution time: training step ×4.3, sampling step ×3.8; and the following approximate increases in memory consumption: training step ×4.4, sampling step ×4.6. These estimates were obtained on a single GPU NVIDIA Tesla V100 32 GB with 4 CPU cores Intel Xeon Gold 6152 (2.1–3.7 GHz). In our experiments, MiAD (Nm = 25) required 4 days for the training procedure (8000 epochs) and 2 hours for the sampling procedure (10 000 crystals) on 2 GPUs NVIDIA Tesla A100 80 GB with 8 CPU cores AMD EPYC 7702 2–3.35 GHz. The training and sampling procedures can also be conducted on a single GPU, though this will incur increased time costs.”
> >
> > We recognize that the reference from the main text to this appendix content is not sufficiently prominent; we will make this pointer clearer in a future revision.

---

> > > ### Author Response · Authors · 2025-11-25
> > >
> > > #### Scalability to larger datasets and efficiency at scale
> > >
> > > - Weakness: “Scalability on larger datasets (e.g., MPTS-52) ...”
> > > - Questions: “How scalable is MiAD to larger and more complex datasets such as MPTS-52?” and “When incorporating mirage atoms at larger scales, how computationally efficient is the model, and are any optimization strategies employed to maintain tractable training and inference?”
> > >
> > > There are several challenges in conducting comparisons on MPTS-52. To our knowledge, prior works have not reported S.U.N. metrics for generative models on MPTS-52, and therefore we did not include such experiments in the initial version of the paper. Metrics such as Structure Validity, Compositional Validity, Coverage (COV-R, COV-P), and distributional distances (e.g., the Wasserstein distance of scalar material properties such as density or number of elements) proposed in Xie et al. (2022) have significant limitations, which we discuss in the appendix Additional metrics, page 19. Nevertheless, we are able to compare the original DiffCSP with MiAD to provide additional evidence of the effectiveness of the proposed methodology.
> > >
> > > It is important to consider the DiffCSP backbone architecture, which operates most effectively with crystals containing fewer than 20 atoms. Applying DiffCSP to MPTS-52 places the model in a regime where performance degrades due to the larger number of atoms. Applying mirage infusion further increases the number of atoms; thus, when we use the same network architecture, that challenge is amplified.
> > >
> > > We trained the original DiffCSP and MiAD on MPTS-52 for 6k epochs (six times the number of epochs used in the original DiffCSP for MPTS-52) and selected the best checkpoints by S.U.N. We use the same neural network as in experiments on MP-20. We did not tune MiAD for MPTS-52, but instead used the configuration identified as optimal on MP-20:
> > >
> > > -> For MP-20: $N_{\text{m}}$ = maximum\_number\_of\_atoms\_in\_mp20 + 5 = 20 + 5 = 25
> > >
> > > -> For MPTS-52: $N_{\text{m}}$ = maximum\_number\_of\_atoms\_in\_mpts52 + 5 = 52 + 5 = 57
> > >
> > > The results are as follows (S - Stability rate $E^{hull} < 0$, UN - Unique and Novel among Stable rate):
> > >
> > > | Name | S (eq-V2) | UN | S.U.N. (eq-V2) | S (CHGNet) | UN | S.U.N. (CHGNet)|
> > > |---|---|---|---|---|---|---|
> > > | DiffCSP | 4.27 | 37.24 | 1.59 | 11.76 | 69.90 | 8.22 |
> > > | MiAD | 5.72 | 55.59 | 3.18 | 13.04 | 76.15 | 9.93 |
> > >
> > > Here, we observe consistent improvements with mirage infusion using the same configuration (without finding optimal hyperparameters for dataset).
> > >
> > > Computational overhead:
> > >
> > > -> 1 epoch of DiffCSP training on MPTS-52: 35 sec
> > >
> > > -> 1 epoch of MiAD ($N_{\text{m}} = 57$) training on MPTS-52: 79 sec ($\approx 2.26$x compared to DiffCSP)
> > >
> > > The overhead is smaller than in MP-20 (appendix Experimental details, paragraph Computational costs, page 19) because the network is already near its upper limit of effective atomic interactions (20 atoms); further increases in atom count then lead to approximately linear, not quadratic, scaling.
> > >
> > > Applying mirage infusion to datasets with a broad range of atom counts is a plausible use case in practical applications. For such datasets, we can vary the number of mirage atoms added per structure while keeping the rest of the procedure unchanged. We demonstrate this variant on MPTS-52.
> > >
> > > Specifically, we set $N_{\text{m}}$ per crystal to $N_{\text{atoms}} + \text{Uniform}[0, 10]$, i.e., each crystal receives 0–10 mirage atoms (+5 on average). At generation start, we sample the number of atoms from the training-set distribution (as in the original DiffCSP), and add a sample from Uniform$[0, 10]$. The results for this variant (MiAD-WRNA: MiAD for Wide Ranges of Number of Atoms) are (S - Stability rate $E^{hull} < 0$, UN - Unique and Novel among Stable rate):
> > >
> > > | Name | S (eq-V2) | UN | S.U.N. (eq-V2) | S (CHGNet) | UN | S.U.N. (CHGNet)|
> > > |---|---|---|---|---|---|---|
> > > | DiffCSP | 4.27 | 37.24 | 1.59 | 11.76 | 69.90 | 8.22 |
> > > | MiAD | 5.72 | 55.59 | 3.18 | 13.04 | 76.15 | 9.93 |
> > > | MiAD-WRNA | 5.55 | 44.14 | 2.45 | 13.03 | 70.91 | 9.24 |
> > >
> > > MiAD-WRNA still provides substantial gains over DiffCSP, although it performs slightly worse than the original MiAD with fixed $N_{\text{m}}$. Its main advantage lies in computational cost:
> > >
> > > -> 1 epoch of DiffCSP training on MPTS-52: 35 sec
> > >
> > > -> 1 epoch of MiAD ($N_{\text{m}} = 57$) training on MPTS-52: 79 sec ($\approx 2.26$x vs DiffCSP)
> > >
> > > -> 1 epoch of MiAD-WRNA (+5 avg) training on MPTS-52: 40 s ($\approx 1.14$x vs DiffCSP)
> > >
> > > Thus, MiAD-WRNA offers a practical trade-off between computational cost and quality. These MPTS-52 experiments, together with Table 3, support the effectiveness of mirage atoms across different strategies for incorporating them into crystals. We will include these results in future revisions of the paper.

---

> > > > ### Author Response · Authors · 2025-11-25
> > > >
> > > > #### Baseline coverage and S.U.N. comparisons
> > > >
> > > > - Comment: “For the S.U.N. comparisons, several important baseline models—such as UniMat[1], TGDMat[2], CrysBFN[3], and Crystal-Text-LLM[4]—are not included.”
> > > >
> > > > -> UniMat [1]: Reports Stability and Novelty but does not specify key details such as the use of StructureMatcher or the total number of generated crystals. We therefore cannot extract a compatible S.U.N. value for comparison.
> > > >
> > > > -> TGDMat [2], CrysBFN [3]: Do not report S.U.N. results in any form.
> > > >
> > > > -> Crystal-Text-LLM [4]: Reports Stability and Novelty separately; as a result, the rate of generating crystals that are both Stable and Novel (S.U.N.) is not available. Optimizing these criteria separately is not equivalent to optimizing their conjunction.
> > > >
> > > > - Comment: “Additionally, key recent baselines like DiffCSP++ and SymmCD are missing from Table 1, while FlowLLM and WyFormer are absent from Table 2.”
> > > >
> > > > -> DiffCSP++ and SymmCD: For these models, S.U.N. was reported with Stability computed via MLIPs. Including them in Table 1 (which is based on DFT-evaluated Stability) would require running new DFT calculations for their generations, which is computationally prohibitive. Even original authors often lack the resources to recompute this metric. We had resources for a single DFT-based S.U.N. computation and used it for the final version of our model. Other entries in Table 1 were taken from the literature.
> > > >
> > > > -> FlowLLM and WyFormer: Table 1 targets the most precise S.U.N. evaluation, where Stability is computed via expensive DFT calculations. Adding further S.U.N. results computed via less accurate MLIPs is not necessary for that table’s purpose.
> > > >
> > > > - Comment: “The omission of these models leads to incomplete and inconsistent comparisons, making it difficult to accurately assess the relative performance and claimed improvements of the proposed approach.”
> > > >
> > > > We agree that reproducing all existing models and evaluating S.U.N. via DFT for each would be the fairest and most consistent comparison. However, this is infeasible due to the number of models, lack of released generations/checkpoints, and in some cases the absence of open-source code. Moreover, given known issues with some reported metrics, a full reproduction effort would be unlikely to be resource-efficient. Within these constraints, we aimed for the broadest and most reliable comparison possible:
> > > >
> > > > -> We conducted one DFT-based S.U.N. evaluation for the most precise comparison against models that report the same metric type.
> > > >
> > > > -> For models that report S.U.N. via CHGNet, we evaluated using the same S.U.N. definition. Due to the limitations of such estimates, we also reproduced as many models as possible and computed S.U.N. using the most accurate and expensive available MLIP, eq-V2.
> > > >
> > > > We endeavored to use only reliable metrics and to perform as broad a comparison as our resources allowed. We hope this clarifies both the difficulty of exhaustive comparisons in this area and the rigor of our experimental setup.
> > > >
> > > > #### Applicability beyond diffusion (LLMs and flows)
> > > >
> > > > - Question: “Since MiAD’s architecture closely follows DiffCSP, apart from diffusion models have you tested it on other frameworks like LLMs or Flow-based models? Do that show similar performance improvement?”
> > > >
> > > > We did not test LLMs because, given their autoregressive nature, they can vary the number of atoms by design; thus, our approach is not particularly motivated in that setting. In our experiments, flow-based models perform at approximately the same level of quality as the original DiffCSP. We did not observe significant differences between them. Given the structural similarity of these models, we expect similar performance after applying mirage infusion. We chose DiffCSP for experiments because, despite being well established, it performs on par in S.U.N. terms with newer generative models that modify the nature of each diffusion component.

---

> > > > > ### Comment · Reviewer_enPF · 2025-11-26
> > > > >
> > > > > Thank you to the authors for the swift and detailed response. That said, several important concerns remain insufficiently addressed.
> > > > >
> > > > > - The authors reiterate that their method is not a mere generalization of DiffCSP, which brings me back to my initial question: Is the proposed framework agnostic to the choice of generative backbone for material generation? Beyond diffusion models, have the authors evaluated the approach with other types of generative frameworks such as Bayesian Flow Networks or standard flow-based models? While the rebuttal notes that flow-based models perform at approximately the same level as the original DiffCSP, the absence of architectural, training, and quantitative details makes it difficult to verify whether the proposed method provides consistent improvements across model families, or whether its benefits are specific to diffusion-based designs.
> > > > >
> > > > > - The qualitative visualizations of the mirage atoms are appreciated, but they should be included directly within the revised manuscript or supplementary materials instead of being hosted externally. This will improve accessibility and clarity for readers.
> > > > >
> > > > > - For larger datasets such as MPTS-52, although the method shows performance improvements, MiAD requires roughly 2.25× more time per epoch compared to DiffCSP due to the added mirage atoms. This raises concerns about the scalability of the approach. For a method intended to scale to increasingly large and complex material datasets, such computational overhead should be either strongly justified by proportionally larger performance gains or addressed through optimization strategies. As it stands, the scalability advantage of the proposed method remains unclear.

---

> > > > > > ### Author Response · Authors · 2025-11-27
> > > > > >
> > > > > > Thank you for the thoughtful and constructive follow-up. We address your three remaining concerns in turn: (i) the generality of our framework across generative backbones, (ii) the placement and accessibility of qualitative visualizations, and (iii) the scalability implications of the additional computational cost introduced by mirage infusion.
> > > > > >
> > > > > > 1) On model-agnosticism beyond diffusion models
> > > > > >
> > > > > > We have not yet applied the proposed method to other classes of generative models such as flow-matching, standard flow-based, or Bayesian flow frameworks. In our experiments, we demonstrate that DiffCSP [1] with mirage infusion achieves state-of-the-art results. Other models that operate on the same crystal representation $(L, F, A)$ - including FlowMM [2], CrystalFlow [3], and CrysBFN [4] - differ primarily in the dynamics defined for particular components of this representation. Consequently, there are, in principle, no constraints that would prevent applying mirage infusion to these models.
> > > > > >
> > > > > > For clarity, our contributions (page 2, lines 65-72) are:
> > > > > > - We propose a simple yet powerful technique, mirage infusion, which broadens the original
> > > > > > space of 3D crystal structures, enabling the diffusion model to modify the number of atoms
> > > > > > in the crystal during the generation process.
> > > > > > - We examine the sensitive parameters of the proposed technique and their impact on the
> > > > > > quality of the generative model through a series of experiments.
> > > > > > - We demonstrate that the proposed approach significantly enhances the performance of the
> > > > > > base joint diffusion model and substantially surpasses the previous state-of-the-art model.
> > > > > >
> > > > > > We have not claimed improvements for all generative models using this representation; accordingly, we did not conduct experiments designed to substantiate such a universal claim. We agree that extending the evaluation to additional backbones would be a valuable addition, and we will include such experiments in the camera-ready version.
> > > > > >
> > > > > > 2) On qualitative visualizations of mirage atoms
> > > > > >
> > > > > > We have incorporated the qualitative visualizations directly into the revised manuscript. They appear in the appendix Qualitative analysis of mirage infusion dynamics, page 22.
> > > > > >
> > > > > > 3) On scalability and computational overhead
> > > > > >
> > > > > > We offer several clarifications regarding the scalability concern:
> > > > > >
> > > > > > - Advancing model quality is orthogonal to reducing computational cost. At present, no alternative method achieves the same performance level as MiAD; thus, mirage infusion is important to the community because it demonstrates how to reach this level of performance.
> > > > > >
> > > > > > - Regarding the request for “proportionally larger performance gains” on MPTS-52: we report the following results for the original MiAD in our previous answer:
> > > > > >   - S.U.N. (eq-V2): DiffCSP 1.59% vs. MiAD 3.18% (≈2× relative improvement over DiffCSP).
> > > > > >   - S.U.N. (CHGNet): DiffCSP 8.22% vs. MiAD 9.93% (≈1.2× relative improvement).
> > > > > >
> > > > > >   On average, MiAD yields approximately a 1.6× quality boost while incurring roughly a 2.25× increase in time per epoch. The observation that these gains are not “proportional” appears to presuppose a linear relationship between quality metrics and computational cost—i.e., a method that is 2.25× more expensive should deliver ≥2.25× improvement. If this is indeed the intended criterion, we would appreciate further justification, as such a linear proportionality is not standard in AI evaluation practice.
> > > > > >
> > > > > > - Regarding the request that this be “addressed through optimization strategies,” we have already proposed an optimized variant of Mirage Infusion, MiAD-WRNA, that is more cost-effective, incurring only 1.14× computational overhead while delivering, on average, a 1.33× improvement in S.U.N. This could be found in our previous answer.
> > > > > >
> > > > > > - With respect to scalability to larger materials datasets: the largest dataset of computationally stable materials currently used in this context, Alex-MP (as in MatterGen [5]), contains around 607k metastable crystals with $E_{\mathrm{hull}} < 0.1$. Even setting aside concerns about dataset quality, synthesizability guarantees, and the practical difficulty of further growing such datasets, materials datasets remain orders of magnitude smaller than those in many other AI domains, and the corresponding models are likewise smaller. In this context, a 2.25× increase in per-epoch training time is, at present, is insignificant.

---

> > > > > > > ### Author Response · Authors · 2025-11-27
> > > > > > >
> > > > > > > [1] Rui Jiao, Wenbing Huang, Peijia Lin, Jiaqi Han, Pin Chen, Yutong Lu, and Yang Liu. Crystal structure prediction by joint equivariant diffusion, 2024a. URL https://arxiv.org/abs/2309.04475.
> > > > > > >
> > > > > > > [2] Benjamin Kurt Miller, Ricky T. Q. Chen, Anuroop Sriram, and Brandon M Wood. Flowmm: Generating materials with riemannian flow matching, 2024. URL https://arxiv.org/abs/2406.04713.
> > > > > > >
> > > > > > > [3] Xiaoshan Luo, Zhenyu Wang, Qingchang Wang, Jian Lv, Lei Wang, Yanchao Wang, and Yanming Ma. Crystalflow: A flow-based generative model for crystalline materials, 2025. URL https://arxiv.org/abs/2412.11693.
> > > > > > >
> > > > > > > [4] Hanlin Wu, Yuxuan Song, Jingjing Gong, Ziyao Cao, Yawen Ouyang, Jianbing Zhang, Hao Zhou, Wei-Ying Ma, and Jingjing Liu. A periodic bayesian flow for material generation, 2025. URL https://arxiv.org/abs/2502.02016.
> > > > > > >
> > > > > > > [5] Claudio Zeni, Robert Pinsler, Daniel Zugner, Andrew Fowler, Matthew Horton, Xiang Fu, Sasha Shysheya, Jonathan Crabbe, Lixin Sun, Jake Smith, Bichlien Nguyen, Hannes Schulz, Sarah Lewis, Chin-Wei Huang, Ziheng Lu, Yichi Zhou, Han Yang, Hongxia Hao, Jielan Li, Ryota Tomioka, and Tian Xie. MatterGen: a generative model for inorganic materials design, 2024. URL https://arxiv.org/abs/2312.03687.

---

### Official Review · Reviewer_zyMX · 2025-10-31

**Soundness:** 3
**Presentation:** 3
**Contribution:** 2
**Rating:** 4
**Confidence:** 3

**Summary:**

The paper focuses on de novo generation of crystalline materials and proposes to use mirage atoms (also called fake or virtual atoms in the literature), i.e. atoms with no type, to increase the model flexibility by allowing the number of atoms to vary during generation. The authors demonstrate that this simple technique leads to better S.U.N rate compared to other de novo generation models.

**Strengths:**

The paper is well written and the idea is simple yet powerful, resulting in a significant improvement of S.U.N rate in the DNG task by keeping the same backbone architecture as some of the baseline model. Additionally, the authors provide ablation studies for all key design choices, with detailed results reported in the appendix. The evaluation of the generated crystalline material samples is conducted both by DFT calculation and machine-learning based interatomic potentials.

**Weaknesses:**

I have the feeling that the contribution may be somewhat limited, even though it appears to provide a benefit in terms of the S.U.N. rate. In addition to [1] there is another concurrent work [2] (also out in August)  that uses virtual or fake nodes for molecules to allow for variable-sized output. In this paper, the idea is applied to materials, although it is not material-specific and could, in principle, be extended to any graph generation problem. As noted in the paper, the authors mask the mirage atom from the loss computation for the fractional coordinates, which they found to beneficial for material generation. It would have been interesting to see whether the same effect occurs in molecular generation. Additionally, the computation of metrics on the generated samples could be explained in greater detail (see Question)

**References**

[1] "Multi-domain Distribution Learning for De Novo Drug Design", Schneuing et al, 2025

[2] "FlowMol3: Flow Matching for 3D De Novo Small-Molecule Generation", Ian Dunn and David Koes, 2025

**Questions:**

- Why not test the way you used fake nodes for crystals on molecules also?
- I am somewhat unsure whether the comparison presented is truly apples-to-apples. All the baseline models sample according to the empirical distribution of the dataset (maximum 20 atoms), while your approach samples up to 25 atoms due to the mirage atoms approach. A potentially a more fair comparison might involve using the empirical distribution generated by your model for the fixed-size baselines (Figure 4 you are presenting in Appendix). Related to the metrics computation, did you exclude generated samples that contain more than 20 atoms in the case of MP-20? Are these generated crystals automatically counted as novel in the evaluation? Since the fixed-size models are limited to the empirical distribution of MP-20, they cannot generate crystals with more than 20 atoms and therefore cannot outperform the proposed method.
- In the Appendix, Table 5 mentions that you attempt to position the mirage atoms at the geometric center of the real atoms. How is this computed, given that the atoms lives in a periodic space? Which geometric mean have you considered?

---

> ### Author Response · Authors · 2025-11-25
>
> We thank the reviewer for their thoughtful assessment and constructive suggestions. We agree that the concept of virtual or mirage atoms is general and can be instantiated in multiple domains of 3D graph generation. Our work focuses on the crystalline materials domain and introduces domain-motivated design choices for the placement of mirage atoms and the associated loss formulation. We provide clarifications regarding cross-domain applicability, fairness of comparisons, and details on metric computation and periodic geometry.
>
> #### General comments on scope and relation to concurrent work
>
> We agree that the concept of virtual atoms is sufficiently general and, in essence, corresponds to introducing virtual components in an object’s representation. Both materials and molecules can be cast as 3D graph generation problems, where this concept reduces to adding virtual nodes. As stated in the paper, we developed our method independently; upon discovering concurrent work in molecular generation, we noted several architectural differences, particularly in (1) how mirage/virtual atoms are positioned and (2) how the loss interacts with these atoms.
>
> Our first design choice is domain-specific. In crystalline materials, we operate on unit cells of finite volume without constraints on where an atom may be placed inside the cell; we therefore use uniform placement. By contrast, molecules are bonded structures, and atoms plausibly arise only within certain distances relative to existing atoms. In that context, placing virtual atoms at a center of mass is a reasonable choice; our uniform placement is similarly well motivated for crystals. Empirically, in the crystal domain, placing mirage atoms at the center of mass performs substantially worse than uniform placement (appendix Ablation, Table 5, page 17). Moreover, the effectiveness of the loss design appears to depend on the placement strategy: in Table 5, masked loss outperforms non-masked loss only when mirage atoms are placed uniformly, whereas with center-of-mass placement, a non-masked loss performs better. In summary, while the overarching concept is general, effective use appears to require domain-specific methodology. We developed such methodology for crystals and obtained state-of-the-art results. We likewise hope these ideas will inform molecular generation.
>
> #### Response to specific questions
>
> - “Why not test the way you used fake nodes for crystals on molecules also?”
>
> We are currently less familiar with generative modeling and evaluation protocols in molecular domains. For crystals, assembling, filtering, and implementing appropriate quality criteria required substantial effort. Some prior works in materials generation rely on metrics with substantial limitations, and even among suitable metrics, evaluation pipelines differ across studies. We therefore concentrated on the domain where we have sufficient expertise to conduct rigorous, defensible evaluation, including clearly articulated comparisons. Entering the molecular domain would require a similarly careful effort to understand domain-specific metrics and conventions -- an important direction that we view as part of a broader research program on adding virtual components to 3D objects but beyond the present scope.

---

> > ### Author Response · Authors · 2025-11-25
> >
> > - “I am somewhat unsure whether the comparison presented is truly apples-to-apples...”
> >
> > We appreciate the opportunity to clarify. We did not exclude generated crystals with more than 20 atoms on MP-20, and, due to the dataset’s definition, all such samples are considered novel. However, they may still be non-unique (if already generated by the model) or unstable. As shown in appendix Distribution of number of atoms, Figure 4, page 21, there are fewer than 10 S.U.N. crystals with more than 20 atoms, whereas the total number of S.U.N. crystals is approximately 820 (among all 10000 crystals generated by MiAD). Excluding those few would reduce the S.U.N. rate from about 8.2% to about 8.1%, leaving the comparative conclusions unchanged.
> >
> > - “In the Appendix, Table 5 mentions that you attempt to position the mirage atoms at the geometric center of the real atoms. How is this computed, given that the atoms live in a periodic space? Which geometric mean have you considered?”
> >
> > We compute the geometric center using the atoms within a single unit cell. The motivation in these experiments was to transfer the design choice for virtual atoms from the molecular domain [1] to crystals. In molecules, placing virtual atoms at a central location provides a point not far from the bonded molecular structure; by analogy, in crystals we sought a point that is not far from the atoms in the unit cell to serve as a common position for all mirage atoms.
> >
> > [1] Arne Schneuing, Ilia Igashov, Adrian W. Dobbelstein, Thomas Castiglione, Michael M. Bronstein, and Bruno Correia. Multi-domain distribution learning for de novo drug design. In The Thirteenth International Conference on Learning Representations, 2025. URL https://openreview.net/forum?id=g3VCIM94ke.

---

### Official Review · Reviewer_wv3Y · 2025-11-02

**Soundness:** 2
**Presentation:** 3
**Contribution:** 2
**Rating:** 2
**Confidence:** 4

**Summary:**

This work proposes a material representation that allows the diffusion process to vary the number of atoms during the sampling process.

**Strengths:**

Strengths:
 - The paper is well written and easy to understand. Related works have been discussed properly.
 - The method is generalizable and can be applied to most pre-existing works without much hassle.
 - Results show the improvement in SUN metrics, although the unique and novel metrics are not better than baselines like FlowMM, DiffCSP and WyFormer.

**Weaknesses:**

Weaknesses:
 - This work only introduces a representation for a material that allows the number of atoms to be flexible. No new model architecture or algorithmic variation has been proposed, which limits the novelty of this work.
 - Datasets with bigger and more complicated structures should have been used to highlight the advantages of this approach. Please also include the results of MPTS-52 dataset in the table.
 - As noted in the ablation studies, the method is sensitive to hyperparameters. like the loss scaling and maximum number of atoms in the augmented atom, which means that for different datasets, these parameters will need to be tuned every time.
 - D3PM introduced several types of transition matrices in their work - uniform, gaussian, mask state. The authors have used the uniform transition matrix for mirage diffusion, but have they tried any other types of transitions? How would the model perform if the mirage atoms are treated as masked states in D3PM?
 - There are some missing citations for the works mentioned in the appendix like CrysBFN and TGDMat.
 - Typos and mistakes - line 292 ("stucture" should be "structure"), line 342 ("does not necessary mean" should be "does not necessarily mean")

**Questions:**

Text Guided Diffusion Models need to be discussed.  Please comment on that.

---

> ### Author Response · Authors · 2025-11-25
>
> We thank the reviewer for their careful reading and constructive feedback. Below, we address each comment in turn. Our goal is to clarify our design choices, provide additional empirical evidence (including new results on MPTS-52), and correct presentation issues noted by the reviewer. We also expand on related directions such as text-guided diffusion and transition-matrix variants in D3PM, and we will incorporate the suggested citations and typographical corrections in a revised version.
>
> - “Results show the improvement in SUN metrics, although the unique and novel metrics are not better than baselines like FlowMM, DiffCSP and WyFormer.”
>
> We agree that the unique and novel metrics are lower. However, unique and novel metrics alone do not reflect overall model quality; for instance, a random generative model would typically achieve even higher unique and novel scores. We report these values only to illustrate how each component of S.U.N. contributes to the final composite metric. Therefore uniqueness and novelty should be considered only in conjunction with the stability, resulting in the superior performance of our proposed method.
>
> - “This work only introduces a representation for a material that allows the number of atoms to be flexible. No new model architecture or algorithmic variation has been proposed, which limits the novelty of this work.”
>
> We would like to note that our training and sampling procedures do include modifications. In the training procedure, we employ a different loss function, and in the sampling procedure, the model operates with a variable number of real atoms in crystal structures. The simplicity of the proposed technique does not diminish its significance, as it leads to substantial improvements in generative quality. Moreover, when applied to the established DiffCSP model, our approach outperforms other generative models that focus on developing new architectures or algorithmic variations. From our perspective, the simplicity of the approach is an additional advantage when it yields superior performance.

---

> > ### Author Response · Authors · 2025-11-25
> >
> > - “Datasets with bigger and more complicated structures should have been used to highlight the advantages of this approach. Please also include the results of MPTS-52 dataset in the table.”
> >
> > There are several challenges in conducting comparisons on MPTS-52. To our knowledge, prior works have not reported S.U.N. metrics for generative models on MPTS-52, and therefore we did not include such experiments in the initial version of the paper. Metrics such as Structure Validity, Compositional Validity, Coverage (COV-R, COV-P), and distributional distances (e.g., the Wasserstein distance of scalar material properties such as density or number of elements) proposed in Xie et al. (2022) have significant limitations, which we discuss in the appendix Additional metrics, page 19. Nevertheless, we are able to compare the original DiffCSP with MiAD to provide additional evidence of the effectiveness of the proposed methodology.
> >
> > It is important to consider the DiffCSP backbone architecture, which operates most effectively with crystals containing fewer than 20 atoms. Applying DiffCSP to MPTS-52 places the model in a regime where performance degrades due to the larger number of atoms. Applying mirage infusion further increases the number of atoms; thus, when we use the same network architecture, that challenge is amplified.
> >
> > We trained the original DiffCSP and MiAD on MPTS-52 for 6k epochs (six times the number of epochs used in the original DiffCSP for MPTS-52) and selected the best checkpoints by S.U.N. We use the same neural network as in experiments on MP-20. We did not tune MiAD for MPTS-52, but instead used the configuration identified as optimal on MP-20:
> >
> > -> For MP-20: $N_{\text{m}}$ = maximum\_number\_of\_atoms\_in\_mp20 + 5 = 20 + 5 = 25
> >
> > -> For MPTS-52: $N_{\text{m}}$ = maximum\_number\_of\_atoms\_in\_mpts52 + 5 = 52 + 5 = 57
> >
> > The results are as follows (S - Stability rate $E^{hull} < 0$, UN - Unique and Novel among Stable rate):
> >
> > | Name | S (eq-V2) | UN | S.U.N. (eq-V2) | S (CHGNet) | UN | S.U.N. (CHGNet)|
> > |---|---|---|---|---|---|---|
> > | DiffCSP | 4.27 | 37.24 | 1.59 | 11.76 | 69.90 | 8.22 |
> > | MiAD | 5.72 | 55.59 | 3.18 | 13.04 | 76.15 | 9.93 |
> >
> > Here, we observe consistent improvements with mirage infusion using the same configuration (without finding optimal hyperparameters for dataset).
> >
> > Computational overhead:
> >
> > -> 1 epoch of DiffCSP training on MPTS-52: 35 sec
> >
> > -> 1 epoch of MiAD ($N_{\text{m}} = 57$) training on MPTS-52: 79 sec ($\approx 2.26$x compared to DiffCSP)
> >
> > The overhead is smaller than in MP-20 (appendix Experimental details, paragraph Computational costs, page 19) because the network is already near its upper limit of effective atomic interactions (20 atoms); further increases in atom count then lead to approximately linear, not quadratic, scaling.
> >
> > Applying mirage infusion to datasets with a broad range of atom counts is a plausible use case in practical applications. For such datasets, we can vary the number of mirage atoms added per structure while keeping the rest of the procedure unchanged. We demonstrate this variant on MPTS-52.
> >
> > Specifically, we set $N_{\text{m}}$ per crystal to $N_{\text{atoms}} + \text{Uniform}[0, 10]$, i.e., each crystal receives 0–10 mirage atoms (+5 on average). At generation start, we sample the number of atoms from the training-set distribution (as in the original DiffCSP), and add a sample from Uniform$[0, 10]$. The results for this variant (MiAD-WRNA: MiAD for Wide Ranges of Number of Atoms) are (S - Stability rate $E^{hull} < 0$, UN - Unique and Novel among Stable rate):
> >
> > | Name | S (eq-V2) | UN | S.U.N. (eq-V2) | S (CHGNet) | UN | S.U.N. (CHGNet)|
> > |---|---|---|---|---|---|---|
> > | DiffCSP | 4.27 | 37.24 | 1.59 | 11.76 | 69.90 | 8.22 |
> > | MiAD | 5.72 | 55.59 | 3.18 | 13.04 | 76.15 | 9.93 |
> > | MiAD-WRNA | 5.55 | 44.14 | 2.45 | 13.03 | 70.91 | 9.24 |
> >
> > MiAD-WRNA still provides substantial gains over DiffCSP, although it performs slightly worse than the original MiAD with fixed $N_{\text{m}}$. Its main advantage lies in computational cost:
> >
> > -> 1 epoch of DiffCSP training on MPTS-52: 35 sec
> >
> > -> 1 epoch of MiAD ($N_{\text{m}} = 57$) training on MPTS-52: 79 sec ($\approx 2.26$x vs DiffCSP)
> >
> > -> 1 epoch of MiAD-WRNA (+5 avg) training on MPTS-52: 40 s ($\approx 1.14$x vs DiffCSP)
> >
> > Thus, MiAD-WRNA offers a practical trade-off between computational cost and quality. These MPTS-52 experiments, together with Table 3, support the effectiveness of mirage atoms across different strategies for incorporating them into crystals. We will include these results in future revisions of the paper.

---

> > > ### Author Response · Authors · 2025-11-25
> > >
> > > - “As noted in the ablation studies, the method is sensitive to hyperparameters, like the loss scaling and maximum number of atoms in the augmented atom, which means that for different datasets, these parameters will need to be tuned every time.”
> > >
> > > First, the proposed method is not sensitive to loss scaling per se. Loss scaling is a general issue for joint diffusion/flow models where the final loss is a sum of losses across multiple diffusion components (e.g., DiffCSP, FlowMM, CrystalFlow, etc.). We included the loss scaling ablation to demonstrate that our performance gains are not due to more favorable loss scaling.
> > >
> > > Second, the ablation on the maximum number of atoms indicates the opposite of sensitivity. One can select $N_{\text{m}}$ slightly larger than the dataset’s maximum atom count -- $+5$ in our experiments -- and still obtain strong performance. The ablation shows that even if we keep the same maximum number of atoms ($20$), the method remains much more effective than the original DiffCSP (following results for DiffCSP were taken from section Experiments, Table 2, page 8, ablation results were taken from appendix Ablation, Table 3, page 16):
> > >
> > > -> DiffCSP: S.U.N. (eq-V2) = 2.5\%, S.U.N. (CHGNet) = 8.9\%
> > >
> > > -> MiAD (20): S.U.N. (eq-V2) = 5.3\%, S.U.N. (CHGNet) = 12.0\%
> > >
> > > -> MiAD (25): S.U.N. (eq-V2) = 5.5\%, S.U.N. (CHGNet) = 12.9\%
> > >
> > > Furthermore, even when $N_{\text{m}}$ exceeds the regime that the DiffCSP architecture handles most efficiently (20 atoms), MiAD still outperforms DiffCSP:
> > >
> > > -> DiffCSP: S.U.N. (eq-V2) = 2.5\%, S.U.N. (CHGNet) = 8.9\%
> > >
> > > -> MiAD (30): S.U.N. (eq-V2) = 4.7\%, S.U.N. (CHGNet) = 11.3\%
> > >
> > > -> MiAD (35): S.U.N. (eq-V2) = 4.6\%, S.U.N. (CHGNet) = 11.8\%
> > >
> > > We respectfully disagree that the method is sensitive; rather, it remains effective even under non-ideal configurations.
> > >
> > > - “D3PM introduced several types of transition matrices in their work -- uniform, Gaussian, mask state. The authors have used the uniform transition matrix for mirage diffusion, but have they tried any other types of transitions? How would the model perform if the mirage atoms are treated as masked states in D3PM?”
> > >
> > > Treating mirage atoms as masked states within D3PM would produce a different trajectory for the number of mirage atoms during generation. In our current implementation, the number of mirage atoms tends to increase during generation. By contrast, with masked-state D3PM, generation typically begins with all tokens masked and then progressively reveals them, decreasing the number of masked states over time. However, by design, masked-state D3PM terminates with no masked states. Consequently, the final crystals would contain no mirage atoms, implying a fixed number of atoms and precluding changes in atom count. More broadly, innovations in diffusion processes for atom types (as well as for lattices or fractional coordinates) are orthogonal to our contribution, and advances in that direction could be combined with our method.
> > >
> > > - “There are some missing citations for the works mentioned in the appendix like CrysBFN and TGDMat.
> > > Typos and mistakes - line 292 (‘stucture’ should be ‘structure’), line 342 (‘does not necessary mean’ should be ‘does not necessarily mean’).”
> > >
> > > We appreciate these comments and will correct the typographical errors in a future revision. We will also add the missing citations for CrysBFN and TGDMat in the Related Work section of the main text.
> > >
> > >
> > > - “Text Guided Diffusion Models need to be discussed. Please comment on that.”
> > >
> > > To the best of our knowledge, TGDMat is the only text-guided diffusion model for crystal generation. It represents an important step toward adapting diffusion models for real-world, conditional generation scenarios. Our work focuses on improving unconditional generation, which is why we did not discuss conditional results in the main text. Nevertheless, our improvements in unconditional generation can be leveraged to extend the base model to conditional generation, as demonstrated in TGDMat and MatterGen.

---

### Author Response · Authors · 2025-12-02
**General Comment for AC**

Dear Area Chair,

We would like to restate our position and summarize what we clarified during the discussion. Our main point is that the method we propose is simple but has a surprisingly strong practical impact on crystalline materials generation. It directly addresses issues that currently limit how generative models work with crystal structures.

__1. Practical gains against many models__

We show consistent improvements in combination of stability, uniqueness and novelty of generated crystals, beating a wide range of leading models on the standard MP-20 benchmark, using both traditional evaluations and the newer eqV2 relaxation pipeline.

__2. Robustness and avoiding known failure modes__

Additional experiments demonstrate that the method is robust to hyperparameter choices and doesn’t struggle from mode collapse toward specific atom counts or space groups.

__3. Scalability to larger systems__

Reviewers raised the question of whether the approach scales to crystals with more atoms. Since MPTS-52 is not a standard benchmark for de novo generation, we did not originally include those experiments. But we ran them during the discussion, since this point mattered to several reviewers.
Importantly, the method works out-of-the-box and still provides substantial improvements over DiffCSP on MPTS-52. These results are now in Appendix K (p. 24) and directly address the concern about scalability.

__4. Qualitative illustration of the mechanism__

At a reviewer’s request, we included an example showing how the method helps the model correct unstable or asymmetric structures, producing more realistic crystals. This is now in Appendix J (p. 22).

__5. Other concerns__

We answered the remaining points in the discussion thread, where the details are easier to follow alongside the submission.

We believe that the combination of simplicity, broad applicability, and demonstrated scalability makes this work a strong candidate for ICLR, especially given the growing interest in reliable generative modeling for scientific discovery.

---

### Meta-Review · Area_Chair_K2XA · 2025-12-27

**Summary:**

This paper introduces mirage infusion, a simple modification to equivariant joint diffusion for crystals that adds “mirage” (non-existent) atoms so the model can change atom count during sampling, addressing a key limitation of fixed-size diffusion generators. Reviewers agree the paper is clearly written and the idea is elegant; results show substantial S.U.N. improvements on MP-20 with both DFT- and MLIP-based stability evaluation. However, concerns were raised about limited methodological novelty, fairness and completeness of comparisons (especially the impact of generating >20-atom structures on MP-20 and missing baselines/metrics inconsistencies), and scalability and compute overhead. The rebuttal added/clarified experiments and provided efficiency numbers and qualitative evidence of an error-correction mechanism, partially addressing several concerns but the rest of the concerns might remain.

**Reviewer Concerns:**

Partly addressed:
- Added MPTS-52 results and a cheaper MiAD-WRNA variant.
- Clarified >20-atom MP-20 fairness impact is negligible.
- Provided compute overhead vs DiffCSP.
- Added qualitative mechanism (mirage removal can “prune” unstable atoms, improve symmetry/stability).
- Acknowledged missing citations and added brief note on text-guided diffusion context.

Still outstanding:
- Novelty and positioning seem incremental (concurrent and related work around the virtual-node idea.
- Results are not significantly better than certain baselines.
- Backbone generality not demonstrated beyond DiffCSP.

**Reviewer Scores:**

wv3Y: Likely up to borderline reject after MPTS-52/robustness clarifications, but novelty and baselines may still cap score.

zyMX: Fairness/metrics concerns addressed; likely unchanged or slight increase.

enPF: Still concerned about backbone and scalability overhead; likely unchanged.

ixHC: Requests largely addressed (MPTS-52, space groups, overhead); likely unchanged or slight increase.

---

### Decision · Program_Chairs · 2026-01-26

Reject